# ITERATIVE AMORTIZED INFERENCE: UNIFYING IN-CONTEXT LEARNING AND LEARNED OPTIMIZERS

## ABSTRACT

Modern learning systems increasingly rely on amortized learning — the idea of reusing computation or inductive biases shared across tasks to enable rapid generalization to novel problems. This principle spans a range of approaches, including meta-learning, in-context learning, prompt tuning, learned optimizers and more. While motivated by similar goals, these approaches differ in how they encode and leverage task-specific information, often provided as in-context examples. In this work, we propose a unified framework which describes how such methods differ primarily in the aspects of learning they amortize — such as initializations, learned updates, or predictive mappings — and how they incorporate task data at inference. We introduce a taxonomy that categorizes amortized models into parametric, implicit, and explicit regimes, based on whether task adaptation is externalized, internalized, or jointly modeled. Building on this view, we identify a key limitation in current approaches: most methods struggle to scale to large datasets because their capacity to process task data at inference (e.g., context length) is often limited. To address this, we propose *iterative amortized inference*, a class of models that refine solutions step-by-step over mini-batches, drawing inspiration from stochastic optimization. Our formulation bridges optimization-based meta-learning with forward-pass amortization in models like LLMs, offering a scalable and extensible foundation for general-purpose task adaptation.

## 1 INTRODUCTION

Consider the problem of modeling the motion of an object on various planets, under different gravitational conditions. While each planet has its own gravitational constant, the underlying dynamics, governed by Newtonian physics, remain invariant. One could train a new model from scratch for each planet, but this would ignore the substantial structure shared across tasks. A more sample-efficient approach is to reuse knowledge acquired from other planets and adapt only a small set of task-specific parameters, such as the gravitational constant. In this setting, the shared inductive bias (e.g., the equations of motion) is *amortized* across tasks, enabling fast, local, and sample-efficient adaptation.

This principle serves as the basis for several methods which avoid solving each task from scratch by learning mechanisms that captures the shared information between tasks, enabling rapid adaptation to new ones (Hospedales et al., 2021; Nichol et al., 2018). For instance, gradient-based meta-learners (Finn et al., 2017; Rusu et al., 2018) encode inductive biases through learned *meta* initialization to facilitate downstream task training. In contrast, amortization is implicitly incentivized in large language models (LLMs) by training on diverse contexts, allowing them to solve new tasks at inference by conditioning on observations (ICL; Brown et al., 2020; Dong et al., 2022) or instructions (Lester et al., 2021; Liu et al., 2021). Learned optimizers (Andrychowicz et al., 2016; Metz et al., 2022a;b; Knyazev et al., 2024) instead amortize optimization itself, learning to predict parameter updates of *fixed* models conditioned on gradients.

We introduce a unified framework that encapsulates a broad spectrum of amortized learning methods − meta-learners, learned optimizers, ICL − as particular instances within a common structure, characterized by two primary components: (1) a shared mechanism to encode task-invariant inductive biases, and (2) a task adaptation function that utilizes data from novel tasks to model task-specific behavior. Our framework identifies three distinct amortization regimes distinguished by the inductive biases involved within each component − (a) *parametric amortization* where a learned function maps task-specific data to corresponding parameters of a fixed model, *e.g.* learned optimizers

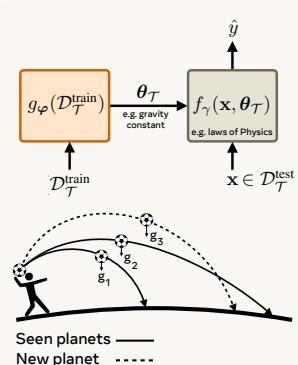

| Method | $g_{\boldsymbol{\varphi}}(\mathcal{D}_{\mathcal{T}}^{\text{train}})$ | $\boldsymbol{\theta}_{\mathcal{T}}$ | $f_{\gamma}(\mathbf{x}, \boldsymbol{\theta}_{\mathcal{T}})$ | Amortization |
|---|---|---|---|---|
| Standard training | SGD | weights | Fixed architecture ($\gamma = \varnothing$) | − |
| MAML | SGD, learned $\boldsymbol{\theta}_0$ | weights | Fixed architecture ($\gamma = \varnothing$) | $\boldsymbol{\theta}_0$ |
| Learned optimizers | Learned updates | weights | Fixed architecture ($\gamma = \varnothing$) | $g_{\boldsymbol{\varphi}}$ |
| ICL | Identity Map | Tokens | Transformer (weights $\gamma$) | $f_{\gamma}$ |
| *Parametric* | Learned updates | weights | Fixed architecture ($\gamma = \varnothing$) | $g_{\boldsymbol{\varphi}}$ |
| *Implicit* | Subsampling | Batch $\mathcal{B}_{\mathcal{T}}^{\text{train}}$ | Transformer (weights $\gamma$) | $f_{\gamma}$ |
| *Explicit* | Learned updates | latents | Transformer (weights $\gamma$) | $g_{\boldsymbol{\varphi}}, f_{\gamma}$ |

Table 1: **Functional decomposition of amortized learners.** We express each method in terms of $g_{\boldsymbol{\varphi}}$ that maps task observations $\mathcal{D}_{\mathcal{T}}^{\text{train}}$ to some representation $\boldsymbol{\theta}_{\mathcal{T}}$ − weights, prompts, dataset itself − which is then fed along with query to $f_{\gamma}(\mathbf{x}, \boldsymbol{\theta}_{\mathcal{T}})$ for prediction. Differences between methods arise from the aspects of learning they amortize, and our proposed taxonomy offers a categorization.

(Andrychowicz et al., 2016) and hypernetworks (Ha et al., 2016), (b) *implicit amortization* where a single model jointly internalizes task-invariant mechanisms and task adaptation through forward-pass conditioning (Brown et al., 2020; Müller et al., 2021), *e.g.* ICL, and (c) *explicit amortization* which disentangles generalization and local adaptation by learning both a dataset-level embedding and a task-conditioned prediction function (Garnelo et al., 2018a; Mittal et al., 2024) via architectural inductive biases. A formal description of these algorithmic components is developed in Section 2, culminating in a central expression in Eq. (5).

These paradigms reflects trade-offs in expressivity, scalability, and efficiency − using gradients, like in learned optimizers, provides direct access to task-specific loss landscapes, making optimization efficient but potentially limited in the richness of task information encoded, while conditioning on observations, like in in-context learning and hypernetworks, allows richer task representations but can be computationally expensive, especially as the conditioning dataset grows. In contrast, pooling-based methods offer compact representations but struggle with fine-grained task variability.

We also propose an iterative amortization framework bridging these perspectives by embracing the stochastic optimization viewpoint. Rather than collapsing task information into a single pass or summary, we model amortization as an iterative refinement process over stochastic mini-batches of task data, leveraging either observations directly or through gradients. This mirrors the success of stochastic gradient descent in scaling optimization to large datasets. By doing so, we generalize the notion of learned optimizers and hypernetworks: instead of operating solely on gradients or generating parameters one-shot, the adaptation function now incorporates streams of mini-batches of observations and corresponding gradients from the task data, enabling greater flexibility and scalability. Our contributions are −

- *Unified Framework*: A general formulation of amortized learning that connects meta-learning, in-context learning, prompt tuning, and learned optimizers as special cases (Section 2).
- *Amortization Taxonomy*: A clear categorization of amortization methods into parametric, implicit, and explicit regimes, based on how they encode inductive bias and adapt to new tasks (Section 3).
- *Stochastic Iterative Amortization*: A scalable approach through iterative refinement over mini-batches, overcoming limitations of scaling to large datasets (Section 4).

## 2 UNIFIED PERSPECTIVE

Given a task $\mathcal{T}$ and a corresponding data distribution $p_{\mathcal{T}}$, a fundamental problem of machine learning is to learn a model $f(\cdot, \boldsymbol{\theta}_{\mathcal{T}}^*)$ that minimizes the true risk based on a loss function $\mathcal{L}$, *i.e.*

$$\boldsymbol{\theta}_{\mathcal{T}}^* = \arg\min_{\boldsymbol{\theta}} \mathbb{E}_{\mathbf{x}, \mathbf{y} \sim p_{\mathcal{T}}} \left[ \mathcal{L}\left(\mathbf{y}, f(\mathbf{x}, \boldsymbol{\theta})\right) \right], \tag{1}$$

which is often accomplished by empirical risk minimization where the task $\mathcal{T}$ is described through a set of *i.i.d* observations $\mathcal{D}_{\mathcal{T}} = \{(\mathbf{x}, \mathbf{y}) \sim p_{\mathcal{T}}\}_i$ and we learn a proxy to the optimal model as

$$\hat{\boldsymbol{\theta}}_{\mathcal{T}} = \arg\min_{\boldsymbol{\theta}} \sum_{(\mathbf{x}, \mathbf{y}) \in \mathcal{D}_{\mathcal{T}}} \mathcal{L}\left(\mathbf{y}, f(\mathbf{x}, \boldsymbol{\theta})\right), \tag{2}$$

in the hope that $\hat{\boldsymbol{\theta}}_{\mathcal{T}}$ generalizes well to new samples from the same distribution, and thus leads to low true risk modeling similar solutions to the optimal one $\boldsymbol{\theta}_{\mathcal{T}}^*$[1]. Owing to theoretical advances

---

[1]The same extends to generative modeling with $\mathbf{y}$ denoting images / sentences and $\mathbf{x}$ class conditioning.

(Shalev-Shwartz & Ben-David, 2014; Zhang et al., 2016; Neal et al., 2018), the scale of data available (Brown et al., 2020; Achiam et al., 2023), and the architectural models and inductive biases (Bahdanau et al., 2014; Vaswani et al., 2017), we have evidence that this paradigm leads to generalizable models. But what happens when we look at a new task?

While effective for single-task learning, the standard empirical risk minimization approach in Eq. (2) learns a separate model $\hat{\boldsymbol{\theta}}_{\mathcal{T}}$ for each task $\mathcal{T}$, using only its associated dataset $\mathcal{D}_{\mathcal{T}}$. This setup suffers from two key limitations: (a) it lacks the ability to systematically adapt to new tasks, even if they are similar, and (b) it fails to leverage cross-task information when tasks $\mathcal{T}_1$ and $\mathcal{T}_2$ share underlying structure. Data from $\mathcal{T}_2$ could provide valuable inductive signal for learning $\hat{\boldsymbol{\theta}}_{\mathcal{T}_1}$, but is ignored.

Traditional meta-learning approaches *e.g.* Model-Agnostic Meta-Learning (MAML; Finn et al., 2017) alleviate this problem by learning global initialization parameters $\hat{\boldsymbol{\theta}}_0$ which encapsulate all the shareable knowledge across $\mathcal{T}$s such that small amounts of finetuning leads to good models $\boldsymbol{\theta}_{\mathcal{T}}$ for any particular task (Nichol & Schulman, 2018; Nichol et al., 2018; Rajeswaran et al., 2019).

$$\hat{\boldsymbol{\theta}}_0 = \arg\min_{\boldsymbol{\theta}_0} \mathbb{E}_{\mathcal{T}} \mathbb{E}_{\mathbf{x}, \mathbf{y}, \mathcal{D}_{\mathcal{T}} \sim p_{\mathcal{T}}} \left[ \mathcal{L}\left(\mathbf{y}, f(\mathbf{x}, \boldsymbol{\theta}_{\mathcal{T}})\right) \right], \quad \text{where} \quad \boldsymbol{\theta}_{\mathcal{T}} = g_{\boldsymbol{\theta}_0}(\mathcal{D}_{\mathcal{T}}). \tag{3}$$

where $g_{\boldsymbol{\theta}_0}$ is an optimization routine, most commonly stochastic gradient descent, with learnable initialization $\boldsymbol{\theta}_0$. Alternatively, hypernetworks (Li & Liang, 2021; Ha et al., 2016; Gaier & Ha, 2019; Jia et al., 2016; Munkhdalai & Yu, 2017) and learned optimizers (Andrychowicz et al., 2016; Li et al., 2017; Metz et al., 2019; Wichrowska et al., 2017; Metz et al., 2022b; Li & Malik, 2017) instead directly model task-specific parameters as outputs of a learned process, often without a notion of "good initialization". Such methods are also naturally described by Equation 3 where $\boldsymbol{\theta}_0$ is replaced by $\boldsymbol{\varphi}$ and $g_{\boldsymbol{\varphi}}$ describes a sequence model which either takes observations (*e.g.* hypernetworks) or gradients (*e.g.* learned optimizers) as input signal.

Certain meta-learning methods like example-based in-context learning or prior fitted networks instead directly model the conditional predictive distribution (Garg et al., 2022; Müller et al., 2021), *i.e.*

$$\hat{\gamma} = \arg\min_{\gamma} \mathbb{E}_{\mathcal{T}} \mathbb{E}_{\mathbf{x}, \mathbf{y}, \mathcal{D}_{\mathcal{T}}} \left[ \mathcal{L}\left(\mathbf{y}, f_{\gamma}(\mathbf{x}, \mathcal{D}_{\mathcal{T}})\right) \right], \tag{4}$$

where the model does not expose explicit task parameters $\boldsymbol{\theta}_{\mathcal{T}}$ anymore, and conditioning on the task is achieved either through direct conditioning on observations $\mathcal{D}_{\mathcal{T}}$ − Prior Fitted Networks (PFNs) or ICL (Brown et al., 2020; Dong et al., 2022; Garg et al., 2022; Von Oswald et al., 2023; Hollmann et al., 2022) − or its compressed natural language description when using LLMs (Mishra et al., 2021; Efrat & Levy, 2020; Sanh et al., 2021; Wei et al., 2022b;a; Min et al., 2022).

The fundamental approach across all these methods is to learn an amortized model that leverages shared knowledge across tasks for each individual task, and thus can generalize to new ones with similar underlying mechanisms. All the above problems can then be unified as

$$\min_{\gamma, \boldsymbol{\varphi}} \mathbb{E}_{\mathcal{T}} \mathbb{E}_{\mathbf{x}, \mathbf{y}, \mathcal{D}_{\mathcal{T}}} \left[ \mathcal{L}\left(\mathbf{y}, f_{\gamma}\left(\mathbf{x}, g_{\boldsymbol{\varphi}}\left(\mathcal{D}_{\mathcal{T}}\right)\right)\right) \right] \tag{5}$$

where $f_{\gamma}(\mathbf{x}, g_{\boldsymbol{\varphi}}(\mathcal{D}_{\mathcal{T}}))^2$ is an inference procedure encapsulating our modeling assumptions about obtaining predictions for query $\mathbf{x}$ and a set of observations $\mathcal{D}_{\mathcal{T}}$. This general framework unifies supervised learning, meta-learning, in-context learning, and learned optimizers under a common formalism by varying which components − $f_{\gamma}$ or $g_{\boldsymbol{\varphi}}$ − are learned and how.

## 2.1 SPECIFIC CASES OF AMORTIZATION

In this section, we describe how individual methods are special cases of our unified framework. Given a set of tasks $\mathcal{T}$ along with their corresponding training $\mathcal{D}_{\mathcal{T}}^{\text{train}}$ and validation $\mathcal{D}_{\mathcal{T}}^{\text{valid}}$ set, the empirical counterpart of our proposed framework can be described as

$$\min_{\gamma, \boldsymbol{\varphi}} \mathbb{E}_{\mathcal{T}} \left[ \sum_{(\mathbf{x}, \mathbf{y}) \in \mathcal{D}_{\mathcal{T}}^{\text{valid}}} \mathcal{L}\left(\mathbf{y}, f_{\gamma}\left(\mathbf{x}, \boldsymbol{\theta}_{\mathcal{T}}\right)\right) \right], \quad \text{where} \quad \boldsymbol{\theta}_{\mathcal{T}} = g_{\boldsymbol{\varphi}}(\mathcal{D}_{\mathcal{T}}^{\text{train}}) \tag{6}$$

where $f_{\gamma}(\cdot, \boldsymbol{\theta}_{\mathcal{T}})$ denotes a model with shared parameters $\gamma$ and task-specific parameters $\boldsymbol{\theta}_{\mathcal{T}}$ (*e.g.* weights, soft prompts, latent states, or context tokens). The function $g_{\boldsymbol{\varphi}}$, parameterized by $\boldsymbol{\varphi}$,

---

[2] $g_{\boldsymbol{\varphi}}(\mathcal{T})$ describes the general form, while we look at cases where $\mathcal{D}_{\mathcal{T}}$ provides information about $\mathcal{T}$.

represents the inner optimization mechanism that maps the support set $\mathcal{D}_{\mathcal{T}}^{\text{train}}$ for task $\mathcal{T}$ to $\boldsymbol{\theta}_{\mathcal{T}}$. The loss $\mathcal{L}$ is then computed on the query set $\mathcal{D}_{\mathcal{T}}^{\text{valid}}$. By selecting different forms for $f$, partitioning parameters between $\boldsymbol{\varphi}$ and $\gamma$, and varying the adaptation function $g_{\boldsymbol{\varphi}}$, this framework subsumes a wide spectrum of learning paradigms, as outlined in Table 1 and Appendix A. In particular, we obtain

- **Gradient-based supervised learning** when $f$ is fixed $-$ *e.g.* a neural network architecture $-$ and $g$ an optimization procedure, like stochastic gradient descent, solving Eq. (2).
- **MAML** when $f$ is fixed and $g_{\boldsymbol{\varphi}}$ is an optimization procedure with learnable initialization $\boldsymbol{\theta}_0 \in \boldsymbol{\varphi}$.
- **Learned Optimizers / Hypernetworks** when $f$ is fixed and $g_{\boldsymbol{\varphi}}$ is a neural network with learnable parameters $\boldsymbol{\varphi}$. We obtain hypernetworks when $g_{\boldsymbol{\varphi}}$ maps observations to parameters, and learned optimizers when $g_{\boldsymbol{\varphi}}$ is an iterative procedure, *i.e.* $g_{\boldsymbol{\varphi}} := h_{\boldsymbol{\varphi}}^{(k)} \circ \ldots \circ h_{\boldsymbol{\varphi}}^{(1)}$ where $h_{\boldsymbol{\varphi}}^{(i)}$ is a sequence model taking $\boldsymbol{\theta}^{(0)}, \ldots, \boldsymbol{\theta}^{(i-1)}$ and their gradients $\nabla^{(j)} := \nabla_{\boldsymbol{\theta}} \sum_{(\mathbf{x},\mathbf{y}) \in \mathcal{B}^{(j)}} \mathcal{L}(\mathbf{y}, f(\mathbf{x}, \boldsymbol{\theta}))\big|_{\boldsymbol{\theta}^{(j)}}$ as input, such that $\boldsymbol{\theta}_{\mathcal{T}} = \boldsymbol{\theta}^{(k)}$ and $\mathcal{B}^{(j)} \subseteq \mathcal{D}_{\mathcal{T}}$.[3]
- **In-Context Learning** when $g$ samples a set of observations and $f_{\gamma}$ is modeled as a predictive sequence model with the context as observations. If $g$ describes $\mathcal{T}$ in natural language, then we recover the prompt-based ICL $-$ an emergent phenomena from pretraining of LLMs (Brown et al., 2020).

**Intuitive explanation of the unified framework**. The unified framework allows us to talk about various amortized systems through a common language. Intuitively, $g_{\boldsymbol{\varphi}}(\mathcal{D})$ describes an optional bottleneck or dimensionality reduction into either low-level parameters or high-level latents. These are then fed into the prediction network $f_{\gamma}$ with a query $\mathbf{x}$ to provide the corresponding final prediction. It is important to note that both $g_{\boldsymbol{\varphi}}$ and $f_{\gamma}$ can be recurrent in nature, as is the case in our proposed extension. This is highlighted in Table 1 with existing methods mathematically highlighted within this framework in Section 2.1. It is important to note that Eq. (5) defines a clear paradigm of information flow, where $g_{\boldsymbol{\varphi}}$ defines how information from the training dataset can impact downstream prediction, and $f_{\gamma}$ defines the prediction operator. In general, $g_{\boldsymbol{\varphi}}$ can be gradient-based optimization (recurrent; *e.g.* MAML), direct prediction of parameters (one-shot; *e.g.* hypernetworks), or identity mapping and thus no bottleneck (*e.g.* ICL). Similarly, $f_{\gamma}$ is a predictor that is either fixed (parametric), learned but operating on a bottleneck (explicit), or learned without bottleneck (implicit); and similarly it can be one-shot or recurrent.

**Probabilistic Interpretation.** The framework presented in Eq. (5) can be equivalently seen from the lens of a probabilistic model with the meta-optimization objective being maximum likelihood. Here, $p_{\gamma}(\cdot | \mathbf{x}, g_{\boldsymbol{\varphi}}(\mathcal{D}_{\mathcal{T}}))$ defines the probabilistic model with $p_{\gamma}$ being the likelihood distribution, with corresponding parameters $f_{\gamma}(\cdot, \cdot)$. A fixed $f$ (*e.g.* linear mapping $\mathbf{w}^T \mathbf{x}$) implies a fixed form of the likelihood, with $g_{\boldsymbol{\varphi}}$ inferring its parameters ($\mathbf{w}$). On the other hand, one could fix the likelihood family - indexed by $\gamma$ - with its partial parameters - aka latents - inferred using $g_{\boldsymbol{\varphi}}$, or (c) directly model the posterior predictive without a clean decomposition on the likelihood.

**Non-unique decomposition.** Importantly, the functional form $f_{\gamma}(\mathbf{x}, g_{\boldsymbol{\varphi}}(\mathcal{D}_{\mathcal{T}}))$ does not define a unique way to split the computation between $f_{\gamma}$ and $g_{\boldsymbol{\varphi}}$. In principle, any computation performed by $g_{\boldsymbol{\varphi}}$, along with its parameters, can be folded into $f_{\gamma}$, making the decomposition arbitrary. However, the converse is not true as $g_{\boldsymbol{\varphi}}$ does not have access to $\mathbf{x}$. Thus, this separation is still meaningful from an *algorithmic perspective* with the following convention: assign all computations independent of the query $\mathbf{x}$ to $g_{\boldsymbol{\varphi}}$, and leave the query-dependent computations to $f_{\gamma}$. In this view, $f_{\gamma}$ processes the query while accessing task-specific information $\mathcal{D}_{\mathcal{T}}$ only through the structure or constraints imposed by $g_{\boldsymbol{\varphi}}$, for example, pooled representations, gradient information, or natural language.

**Learning the learner.** So far, we only focused on the modeling assumptions $-$ the form of $f_{\gamma}(\mathbf{x}, g_{\boldsymbol{\varphi}}(\mathcal{D}_{\mathcal{T}}))$ $-$ behind different amortized learners did not address how the optimization problem in Eq. (6) is actually performed. The main complexity comes from iterative natures of $f_{\gamma}$ and $g_{\boldsymbol{\varphi}}$ and the presence of higher order gradients if $g_{\boldsymbol{\varphi}}$ describes a gradient-based procedure. Multiple approaches tackle this problem through first-order approximations (Finn et al., 2017; Nichol & Schulman, 2018), back-propagation through time (Ha et al., 2016), reinforcement learning (Andrychowicz et al., 2016) or evolutionary strategies (Metz et al., 2022b) depending on the form of $g_{\boldsymbol{\varphi}}$.

---

[3]A system is Markovian if it only relies on the current $(\boldsymbol{\theta}^{(j)}, \nabla^{(j)})$ or non Markovian if on more past states.

Figure 1: Iterative Amortized Inference for parametric, explicit and implicit parameterizations.

## 3 AMORTIZED LEARNING SYSTEMS: A TAXONOMY

Motivated by the unified formulation of general-purpose amortized models and their ability to efficiently generalize to novel tasks, we introduce a taxonomy categorizing such models into three classes: parametric, explicit, and implicit. While we focus our attention to tasks described through a set of observations $\mathcal{D}_{\mathcal{T}}$ rather than language or task descriptors, the same categorization holds in general.

**Parametric**. We define the class of amortized models with a fixed $f$ and learnable $g_{\varphi}$ as *parametric amortization*. This includes hypernetworks, learned optimizers, and other approaches to parametric inference using ICL. Here, the functional form of the likelihood is fixed, for example a known simulator (Cranmer et al., 2020) or a categorical likelihood with linear mapping, while $g_{\varphi}$ is an inference or estimation procedure that discovers the optimal parameters (e.g., linear coefficients) from the observed data. Several works (Mittal et al., 2025b; Reuter et al., 2025; Chang et al., 2024) approximate the posterior distribution over parameters of the likelihood as $g_{\varphi}$ while others (Mittal et al., 2025a) investigate the interplay between point-based and distribution-based modeling of $g_{\varphi}$[4]. Relying on a fixed parametric assumption enables us to leverage gradient information in addition to $\mathcal{D}_{\mathcal{T}}$ to infer the parameters of $f$. These inferred parameters can also offer a low-dimensional representation of the task and provide interpretability benefits, especially when $f$ possesses specific, interpretable structure.

**Implicit**. In contrast, we refer to amortized models with a trainable $f_{\gamma}$ and a fixed $g$ as *implicit amortization*, which subsumes in-context learning and specific cases of prior fitted networks and conditional neural processes (Nguyen & Grover, 2022; Hollmann et al., 2022). In this setting, a single trained model takes both the query and the set of observations as input and directly learns to model predictions. The function $g$ is typically either the identity mapping or a subsampling mechanism used to manage large datasets. Unlike parametric approaches, the functional form of the output is learned directly, bypassing explicitly inferring any task-specific parameters. It is non-trivial to pinpoint what part of network activations correspond to dataset-specific vs prediction based activations. Some works (Von Oswald et al., 2023; Nichani et al., 2024) demonstrate that, under certain assumptions, implicit models recover algorithmic behaviors such as gradient descent or causal discovery. However, it remains unclear how these findings generalize beyond specific setups, apart from the broader perspective of learning the posterior predictive distribution (Müller et al., 2021; Garg et al., 2022; Fifty et al., 2023; Vettoruzzo et al., 2024).

**Explicit**. It is possible to leverage the dimensionality reduction benefits offered by the parametric approach while still utilizing a learnable form for the likelihood which the implicit model affords. This is accomplished with a trainable $f_{\gamma}$, which learns the likelihood form shared across tasks like the laws of physics, as well as $g_{\varphi}$, which provides a *low-dimensional* encapsulation of the entire dataset like the gravitational constant. The explicit models considered in (Mittal et al., 2024; Elmoznino et al., 2024) and neural processes (Garnelo et al., 2018a;b) that first learn an embedding of the dataset can be seen as specific cases of this approach. Similar to parametric methods, it allows using gradient information and provides dimensionality reduction and interpretability benefits; but at the cost of inherent non-stationarity during learning $- f_{\gamma}$ and $g_{\varphi}$ are dependent on each other but have to be learned together.

## 4 ITERATIVE AMORTIZED INFERENCE

A fundamental limitation of the existing approaches is their inability to leverage large-scale dataset as conditioning $-$ they are either restricted by context length or rely solely on low-dimensional pooling operations or gradients which can be quite restrictive. Analogously, non-amortized learners like gradi-

---

[4]In general $g_{\varphi}$ can represent probability distributions or sampling operations.

| Training Tasks → | | | MNIST | | FMNIST | | ImageNet | | |
|---|---|---|---|---|---|---|---|---|---|
| Signal ↓ | Steps | Lin Reg | MNIST | FMNIST | FMNIST | MNIST | ImageNet | CIFAR10 | CIFAR100 |
| *Grad* | 1 | $51.3_{\pm 0.3}$ | $76.2_{\pm 2.0}$ | $66.7_{\pm 1.0}$ | $63.8_{\pm 1.7}$ | $79.9_{\pm 1.6}$ | $88.5_{\pm 0.1}$ | $44.9_{\pm 4.5}$ | $92.5_{\pm 1.1}$ |
| | 5 | $4.1_{\pm 0.1}$ | $48.6_{\pm 1.2}$ | $51.8_{\pm 0.5}$ | $41.6_{\pm 1.5}$ | $49.5_{\pm 3.0}$ | $83.1_{\pm 0.2}$ | $17.9_{\pm 1.0}$ | $88.1_{\pm 0.8}$ |
| | 10 | $0.5_{\pm 0.0}$ | $39.7_{\pm 0.7}$ | $43.5_{\pm 0.9}$ | $38.1_{\pm 0.4}$ | $40.7_{\pm 0.8}$ | $83.4_{\pm 0.1}$ | $14.7_{\pm 0.2}$ | $88.1_{\pm 0.6}$ |
| *Data* | 1 | $16.3_{\pm 0.2}$ | $43.9_{\pm 1.3}$ | $40.5_{\pm 1.2}$ | $35.9_{\pm 1.7}$ | $38.8_{\pm 1.4}$ | $90.8_{\pm 0.2}$ | $60.9_{\pm 3.5}$ | $93.9_{\pm 1.1}$ |
| | 5 | $0.5_{\pm 0.0}$ | $35.5_{\pm 0.5}$ | $34.3_{\pm 0.6}$ | $30.8_{\pm 0.3}$ | $31.3_{\pm 0.5}$ | $95.2_{\pm 0.1}$ | $46.9_{\pm 1.8}$ | $96.3_{\pm 0.6}$ |
| | 10 | $0.3_{\pm 0.0}$ | $36.6_{\pm 0.3}$ | $35.3_{\pm 0.3}$ | $29.8_{\pm 0.3}$ | $29.1_{\pm 0.4}$ | $93.2_{\pm 0.1}$ | $28.8_{\pm 0.8}$ | $95.0_{\pm 0.8}$ |
| *Grad* | 1 | $25.1_{\pm 0.3}$ | $52.3_{\pm 1.5}$ | $48.2_{\pm 0.8}$ | $37.7_{\pm 1.9}$ | $41.9_{\pm 1.3}$ | $96.7_{\pm 0.1}$ | $70.2_{\pm 4.2}$ | $97.2_{\pm 0.8}$ |
| *+* | 5 | $0.6_{\pm 0.0}$ | $39.2_{\pm 1.0}$ | $37.5_{\pm 0.3}$ | $32.0_{\pm 0.7}$ | $33.4_{\pm 0.5}$ | $93.8_{\pm 0.1}$ | $42.1_{\pm 2.4}$ | $94.8_{\pm 0.7}$ |
| *Data* | 10 | $0.4_{\pm 0.0}$ | $32.0_{\pm 0.5}$ | $32.0_{\pm 0.2}$ | $30.5_{\pm 0.2}$ | $30.3_{\pm 0.3}$ | $92.5_{\pm 0.1}$ | $24.8_{\pm 1.0}$ | $94.1_{\pm 0.4}$ |

Table 2: Our experiments on *parametric amortization* reveal benefits of multiple steps of iterative refinement across ID and OoD (gray columns) evaluation − *lower is better ↓*. Top row describes pre-training tasks and the second row evaluation tasks. We also see that sole reliance on gradients is insufficient and leveraging observations directly can improve performance with fewer iterations.

ent descent resolve this scalability issue through a stochastic mini-batch framework which iteratively refines an approximate solution based on a subset of observations, not the whole dataset. Note that this iterative refinement is Markovian[5] and greedy, *i.e.* the update given current state is independent of history and leads to immediate single-step improvement which could be sub-optimal over multiple steps.

Taking inspiration from SGD and the connection between amortized meta-learners and optimization routines (Elmoznino et al., 2024), we extend existing approaches using an iterative refinement approach; instead of directly modeling predictions, embeddings or parameters, we iteratively refine the output from the previous step greedily using mini-batches as input to a trained sequence model.

For *parametric* and *explicit* amortization, we achieve this by considering $g_{\varphi}$ as an iterative application of a learned sequence model $h_{\varphi}$ on different subsampled training batches $\mathcal{B}_{\text{train}}^{(i)} \subset \mathcal{D}_{\mathcal{T}}^{\text{train}}$, starting from a learned initialization $\boldsymbol{\theta}^{(0)}$.[6] This model $h_{\varphi}$ takes the current state $\boldsymbol{\theta}^{(i)}$ and a mini-batch $\mathcal{B}_{\text{train}}^{(i)}$ as input and returns a refined state $\boldsymbol{\theta}^{(i+1)}$ which decreases validation loss.

$$\boldsymbol{\theta}^{(0)} \xrightarrow{h_{\varphi}(\cdot, \mathcal{B}_{\text{train}}^{(0)})} \boldsymbol{\theta}^{(1)} \xrightarrow{h_{\varphi}(\cdot, \mathcal{B}_{\text{train}}^{(1)})} \dots \xrightarrow{h_{\varphi}(\cdot, \mathcal{B}_{\text{train}}^{(k-1)})} \boldsymbol{\theta}^{(k)} =: \boldsymbol{\theta}_{\mathcal{T}} \tag{7}$$

denotes a $k$-step iterative refinement procedure for parametric and explicit models. While learned optimizers already utilize a mini-batch approach, they only consider parametric cases with fixed $f$ and the information about observations is fed to $h_{\varphi}$ solely through gradients. In particular, they model $h_{\varphi}(\boldsymbol{\theta}, \mathcal{B}) := \text{Transformer}_{\varphi}\left(\boldsymbol{\theta}, \nabla_{\boldsymbol{\theta}} \sum_{(\mathbf{x},\mathbf{y}) \in \mathcal{B}} \mathcal{L}\left(\mathbf{y}, f_{\gamma}\left(\mathbf{x}, \boldsymbol{\theta}\right)\right)\big|_{\boldsymbol{\theta}}\right)$ where the learned model takes a sequence of parameter updates and gradients. Such methods can only recover gradient-based inference routines and cannot model more general optimization schemes (*e.g.* closed-form linear regression solution) as the mapping from observations to gradient is non-invertible.

In parametric and explicit methods, $\boldsymbol{\theta}$s can be seen as recurrent memory compressing useful information of the current task from $\mathcal{D}_{\mathcal{T}}$ to make predictions for new $\mathbf{x}$, *i.e.* prediction for $\mathbf{x}$ given $\boldsymbol{\theta}_{\mathcal{T}}$ is conditionally independent of $\mathcal{D}_{\mathcal{T}}$. In contrast, the *implicit* formulation does not consider task-specific memory as its recurrent state is always tied to the query. In this case, $g_{\varphi}$ simply subsamples mini-batches while $f_{\gamma}$ is a recurrent application of a transformer $r_{\gamma}$ with weights $\gamma$

$$\hat{\mathbf{y}}^{(0)} \xrightarrow{r_{\gamma}([\mathbf{x},\hat{\mathbf{y}}^{(0)}], \mathcal{B}_{\text{train}}^{(0)})} \hat{\mathbf{y}}^{(1)} \xrightarrow{r_{\gamma}([\mathbf{x},\hat{\mathbf{y}}^{(1)}], \mathcal{B}_{\text{train}}^{(1)})} \dots \xrightarrow{r_{\gamma}([\mathbf{x},\hat{\mathbf{y}}^{(k-1)}], \mathcal{B}_{\text{train}}^{(k-1)})} \hat{\mathbf{y}}^{(k)}, \tag{8}$$

where $k$ is the number of iterative refinement steps for implicit models, $[\cdot, \cdot]$ describes a token and $\hat{\mathbf{y}}$ the query-specific predictions which form the recurrent states in this framework and are sequentially updated with $\hat{\mathbf{y}}^{(0)}$ being a learned initialization[7]. Here, $f_{\gamma}$ gets the current prediction state $\hat{\mathbf{y}}^{(i)}$ and a mini-batch $\mathcal{B}_{\text{train}}^{(i)}$ as input and provides a refined prediction $\hat{\mathbf{y}}^{(i+1)}$. Since the implicit model never exposes task specific parameters $\boldsymbol{\theta}_{\mathcal{T}}$, it is non-trivial to leverage gradient information.

---

[5]Certain second-order gradient-based learners are non Markovian, or Markovian in an augmented state space.

[6]In theory, this should learn the appropriate prior for the class of tasks for the parametric setup.

[7]In theory, this initialization should learn the marginal (unconditional) distribution over predictions.

| Training Tasks → | | | MNIST | | FMNIST | |
|---|---|---|---|---|---|---|
| Signal ↓ | Steps | Lin Reg | MNIST | FMNIST | FMNIST | MNIST |
| *Grad* | 1 | $55.5_{\pm0.5}$ | $90.4_{\pm0.7}$ | $90.8_{\pm0.4}$ | $88.2_{\pm0.3}$ | $89.3_{\pm0.3}$ |
| | 5 | $9.4_{\pm0.1}$ | $86.9_{\pm0.1}$ | $84.4_{\pm0.5}$ | $70.8_{\pm0.5}$ | $76.4_{\pm0.4}$ |
| | 10 | $2.9_{\pm0.0}$ | $81.4_{\pm0.5}$ | $78.9_{\pm0.9}$ | $56.4_{\pm0.7}$ | $65.7_{\pm0.4}$ |
| *Data* | 1 | $19.7_{\pm0.4}$ | $89.2_{\pm1.0}$ | $90.0_{\pm0.1}$ | $90.0_{\pm0.1}$ | $90.3_{\pm0.2}$ |
| | 5 | $2.6_{\pm0.0}$ | $88.7_{\pm0.0}$ | $90.0_{\pm0.0}$ | $90.0_{\pm0.0}$ | $90.2_{\pm0.0}$ |
| | 10 | $2.4_{\pm0.0}$ | $88.8_{\pm0.2}$ | $90.0_{\pm0.0}$ | $90.1_{\pm0.1}$ | $90.2_{\pm0.0}$ |
| *Grad* | 1 | $24.4_{\pm0.4}$ | $72.5_{\pm0.7}$ | $62.9_{\pm0.8}$ | $51.1_{\pm1.2}$ | $61.2_{\pm0.6}$ |
| + | 5 | $2.6_{\pm0.0}$ | $67.1_{\pm0.2}$ | $58.3_{\pm0.3}$ | $47.5_{\pm0.3}$ | $57.7_{\pm0.2}$ |
| *Data* | 10 | $2.1_{\pm0.0}$ | $75.1_{\pm0.1}$ | $69.5_{\pm0.1}$ | $45.1_{\pm0.3}$ | $54.4_{\pm0.1}$ |

Table 3: On *explicit amortization* we consistently see improved performance with increasing number of steps (OoD evaluation in gray columns) with error as metric − *lower is better ↓*. Here, we see that reliance on gradients is essential for more complex problems and in particular, explicit models struggled to scale to ImageNet and could only obtain random chance performance.

Motivated by SGD, we learn $\varphi$ and $\gamma$ through a greedy strategy where training is done solely on single-step improvements for all three paradigms. The respective computational graphs are provided below, where we prevent gradients to flow back from $\boldsymbol{\theta}^{(i)}$ or $\hat{\mathbf{y}}^{(i)}$

$$\boldsymbol{\theta}^{(i)} \xrightarrow{h_{\boldsymbol{\varphi}}(\cdot,\mathcal{B})} \boldsymbol{\theta}^{(i+1)} \to \mathcal{L}(\mathbf{y}, f_{\gamma}(\mathbf{x}, \boldsymbol{\theta}^{(i+1)})) \quad \text{or} \quad \hat{\mathbf{y}}^{(i)} \xrightarrow{r_{\gamma}([\mathbf{x},\cdot],\mathcal{B})} \hat{\mathbf{y}}^{(i+1)} \to \mathcal{L}(\mathbf{y}, \hat{\mathbf{y}}^{(i+1)}) \quad (9)$$

Here, an outer loop is used to obtain the states at the $i^{th}$ step − $\boldsymbol{\theta}^{(i)}$ and $\hat{\mathbf{y}}^{(i)}$ respectively. Note that in this setup, $(\mathbf{x}, \mathbf{y}) \in \mathcal{D}_{\mathcal{T}}^{\text{valid}}$ while $\mathcal{B} \subseteq \mathcal{D}_{\mathcal{T}}^{\text{train}}$, *i.e.* we are amortizing a learner to minimize validation loss. We refer to Figure 1 for an illustration of the proposed framework and to Appendix B for a walk-through with an example. For evaluation, we use training and evaluation splits from completely new tasks $\mathcal{T}$ and thus test for generalization to new problems. All three frameworks − *parametric*, *explicit* and *implicit* − modify causal masking to provide efficient and parallelizable computation of $\boldsymbol{\theta}$ and $\hat{\mathbf{y}}$ with mini-batches having varied number of observations, detailed in Appendix E.

**Intuitive explanation of iterative amortized inference.** IAI extends existing one-shot meta learners (*e.g.* ICL, hypernetworks, neural processes) through a recurrent formulation, where either $g_{\varphi}$ or $f_{\gamma}$ from Eq. (5) follows a recurrent structure. This formulation is inspired by stochastic optimization methods, that iteratively update a state based on incoming batches of data. Our dichotomy of parametric, explicit and implicit defines the structure of the state, which for clarity we describe in Table 7. In addition, a clear mathematical description of existing methods and proposed dichotomy is present in Table 1.

## 5 EXPERIMENTS

We evaluate our proposed greedy iterative refinement approach as well as the pros and cons of using the two modalities − observations or gradients − through a suite of experiments. We consider a range of predictive and generative modeling tasks listed below and detailed in Appendix C to assess both in-distribution (ID) and out-of-distribution (OoD) performance under challenging task variations. Learned $g_{\varphi}, f_{\gamma}$ are parameterized as transformers with maximum sequence length of 100 observations, following Kirsch et al. (2022) (details in Appendix E).

**Tasks**. Our tasks include

- **Linear Regression**. $\mathcal{T}$ is defined by ground-truth linear coefficients $\mathbf{w}_{\mathcal{T}} \in \mathbb{R}^{100}$ which control the parameters of $p_{\mathcal{T}}(\mathbf{y}|\mathbf{x})$.
- **MNIST / FashionMNIST Classification**. $\mathcal{T}$ involves a random projection $W_{\mathcal{T}} \in \mathbb{R}^{784 \times 100}$ applied to image pixels and a random label mapping (*e.g.* digits 4 are mapped to label 7; even digits grouped together, etc.) which preserves the semantic structure (Kirsch et al., 2022). We also consider OoD settings where models are trained on MNIST and evaluated on FashionMNIST, and vice versa.
- **ImageNet Classification**. $\mathcal{T}$ samples images from 100 different ImageNet classes and randomly regrouping them into a maximum 100-way classification task, similar to above. Evaluation is done on ImageNet validation data as well as OoD tasks like CIFAR-10 and CIFAR-100. The images are embedded using Dino-v2 (Oquab et al., 2023) before feeding to the transformer.

| Training Tasks → | MNIST | | | FMNIST | | ImageNet | | |
|---|---|---|---|---|---|---|---|---|
| Steps | Lin Reg | MNIST | FMNIST | FMNIST | MNIST | ImageNet | CIFAR10 | CIFAR100 |
| 1 | $14.8_{\pm0.1}$ | $24.9_{\pm0.4}$ | $31.9_{\pm0.4}$ | $22.8_{\pm0.1}$ | $29.1_{\pm0.4}$ | $43.0_{\pm0.7}$ | $19.4_{\pm0.5}$ | $73.9_{\pm0.1}$ |
| 5 | $6.2_{\pm0.0}$ | $12.4_{\pm0.1}$ | $26.1_{\pm0.2}$ | $18.8_{\pm0.1}$ | $16.9_{\pm0.2}$ | $13.3_{\pm0.2}$ | $15.0_{\pm0.1}$ | $59.4_{\pm0.2}$ |
| 10 | $4.7_{\pm0.0}$ | $9.7_{\pm0.1}$ | $24.2_{\pm0.2}$ | $17.8_{\pm0.1}$ | $15.7_{\pm0.2}$ | $12.3_{\pm0.1}$ | $15.9_{\pm0.3}$ | $54.2_{\pm0.4}$ |

Table 4: On *implicit amortization* show consistent improvements in performance with increasing number of steps across a wide range of predictive tasks, with error as metric − *lower is better ↓*.

| | 20 Nodes | | | | 50 Nodes | | | |
|---|---|---|---|---|---|---|---|---|
| Steps | LIN IN | RFF IN | LIN OUT | RFF OUT | LIN IN | RFF IN | LIN OUT | RFF OUT |
| 1 | $0.52_{\pm0.05}$ | $0.43_{\pm0.04}$ | $0.64_{\pm0.04}$ | $0.51_{\pm0.06}$ | $0.7_{\pm0.05}$ | $0.66_{\pm0.04}$ | $0.78_{\pm0.03}$ | $0.69_{\pm0.08}$ |
| 5 | $0.52_{\pm0.06}$ | $0.35_{\pm0.05}$ | $0.68_{\pm0.08}$ | $0.33_{\pm0.06}$ | $0.66_{\pm0.05}$ | $0.56_{\pm0.04}$ | $0.74_{\pm0.04}$ | $0.67_{\pm0.05}$ |
| 10 | $0.44_{\pm0.11}$ | $0.37_{\pm0.07}$ | $0.66_{\pm0.08}$ | $0.61_{\pm0.05}$ | $0.64_{\pm0.06}$ | $0.52_{\pm0.04}$ | $0.77_{\pm0.03}$ | $0.73_{\pm0.03}$ |

Table 5: We evaluate implicit models on topological order prediction, where ID uses SCMs with linear (LIN IN) / non-linear (RFF IN) mechanisms for 20 node graphs and OoD (gray columns) changes function parameters (LIN OUT / RFF OUT) or graph size (50 nodes), with classification error as metric − *lower is better ↓*.

- **Topological Order Prediction**. Following (Scetbon et al., 2024), each task involves sampling a structural causal model (SCM) and inferring its topological order using only observational data.
- **Mixture of Gaussian (MoG) Generation**. Each task is defined by a MoG with the number of mixture components and corresponding means sampled randomly per task. We apply the flow-matching framework (Lipman et al., 2022; Tong et al., 2023; Albergo et al., 2023) to learn the conditional vector field given task data $\mathcal{D}_{\mathcal{T}}$, and evaluate samples generated conditioned on novel densities as context unseen during training.
- **Alphabet Generation**. Following (Atanackovic et al., 2024), we use a dataset of alphabets with different scales and rotations. The task is to leverage the conditioning information to draw samples from the underlying density, with OoD settings corresponding to alphabets unseen during training.

**Metrics**. For predictive tasks, we look at $l_2$ loss for regression and error for classification, while for generative modeling problems we consider the 2-Wasserstein ($\mathcal{W}_2$) and 1-Wasserstein ($\mathcal{W}_1$) distance between samples from the true and generated distribution. We refer to Appendix D for details.

**Baselines**. For parametric amortization, we consider transformer-based hypernetworks and learned optimizers as baselines. The former provides single-step amortization using only data, while the latter is a multi-step system that relies solely on gradients as input. For the explicit models, the 1-step approach solely based on data is equivalent to the explicit model defined in Mittal et al. (2024) as well as to conditional neural processes (Garnelo et al., 2018a). Finally, for the implicit setup, the 1-step approach boils down to ICL (Mittal et al., 2024; Garg et al., 2022; Müller et al., 2021; Von Oswald et al., 2023).

## 5.1 ITERATIVE AMORTIZED INFERENCE PROVIDES CONSISTENT IMPROVEMENTS

**Parametric.** We first test our framework on parametric models − $f$ is a linear predictor and $g_{\boldsymbol{\varphi}}$ is trained to infer its parameters $\boldsymbol{\theta}_{\mathcal{T}}$. We use a transformer $g_{\boldsymbol{\varphi}}$ which amortizes by conditioning on $\mathcal{D}_{\mathcal{T}}$, either through gradient signal or observations, or both. Table 2 demonstrates that iterative amortization consistent improves performance with increasing number of steps across various tasks and input signals − *i.e.* gradients, observations, or both, consistently outperforming the baselines. The amortized model also generalizes OoD: for *e.g.* pre-training on ImageNet classification can lead to good performance on CIFAR-10 at inference, based solely on in-context examples.

**Explicit.** Next, we consider the explicit model − $f_{\gamma}$ is a trained MLP taking a query $\mathbf{x}$ and latent $\boldsymbol{\theta}_{\mathcal{T}}$ as input, while $g_{\boldsymbol{\varphi}}$ is a transformer inferring the latent from $\mathcal{D}_{\mathcal{T}}$. Similar to the parametric case, Table 3 showcases consistent improvements of our proposed approach. In contrast to the parametric experiments, we see an increased importance of gradient information in inferring the right latents. Since there is inherent non-stationarity in the optimization procedure we hypothesize that gradients provide a clearer signal for the inference mechanism $g_{\boldsymbol{\varphi}}$.

**Implicit.** Finally, we consider implicit parameterization where $f_{\gamma}$ jointly leverages both training dataset $\mathcal{D}_{\mathcal{T}}$ as well as the query $\mathbf{x}$ to model predictions $\mathbf{y}$. Our experiments in Tables 4 and 5 indicate that iterative amortization substantially outperforms competitive baselines and leads to improved performance with increasing number of steps. In addition to predictive tasks, we see similar trends

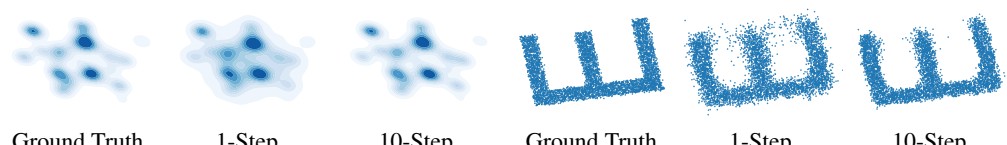

| Ground Truth | 1-Step | 10-Step | Ground Truth | 1-Step | 10-Step |

Figure 2: Samples generated from the *implicit* generative model for GMM and Alphabets task.

| | Alphabets | | | | GMM | | | |
| | ID | | OoD | | 2-dimensional | | 5-dimensional | |
| Steps | $\mathcal{W}_2$ | $\mathcal{W}_1$ | $\mathcal{W}_2$ | $\mathcal{W}_1$ | $\mathcal{W}_2$ | $\mathcal{W}_1$ | $\mathcal{W}_2$ | $\mathcal{W}_1$ |
| --- | --- | --- | --- | --- | --- | --- | --- | --- |
| 1 | $0.28_{\pm 0.00}$ | $0.20_{\pm 0.00}$ | $0.78_{\pm 0.01}$ | $0.64_{\pm 0.00}$ | $1.12_{\pm 0.01}$ | $0.58_{\pm 0.00}$ | $3.27_{\pm 0.01}$ | $1.82_{\pm 0.01}$ |
| 5 | $0.31_{\pm 0.00}$ | $0.22_{\pm 0.00}$ | $0.72_{\pm 0.01}$ | $0.58_{\pm 0.00}$ | $0.90_{\pm 0.01}$ | $0.38_{\pm 0.00}$ | $2.50_{\pm 0.02}$ | $1.05_{\pm 0.01}$ |
| 10 | $0.29_{\pm 0.00}$ | $0.21_{\pm 0.00}$ | $0.67_{\pm 0.01}$ | $0.54_{\pm 0.01}$ | $0.81_{\pm 0.02}$ | $0.32_{\pm 0.01}$ | $2.37_{\pm 0.02}$ | $0.95_{\pm 0.01}$ |

Table 6: We see improved sample quality (under Wasserstein metrics) using *implicit amortization* with increasing number of steps for generative modeling over OoD alphabets and new GMMs − *lower is better* ↓.

on generative modeling where the task is to model the underlying distribution described through the context examples. The amortized model is trained to infer the conditional vector field which interpolates a path between standard normal and the observed distribution, conditioned on $\mathcal{D}_{\mathcal{T}}$ (Atanackovic et al., 2024; Chang et al., 2024). Table 6 and Fig. 2 showcase improved ability of modeling the underlying distributions defined by the context with more steps[8].

We refer to Appendix F for additional comparisons to transformers without causal masking.

## 5.2 ANALYSIS

**Gradient signal is often sub-optimal.** When $g_{\boldsymbol{\varphi}}$ solely leverages gradient information, we recover the space of learned optimizers. For relatively lower dimensional problems, we see that the space of sole gradient-based learned optimizers is suboptimal and additionally leveraging the observations can lead to substantial improvements. Alternatively, in higher dimensions since the hypothesis space is larger, it is harder for the model to infer the right parameters. In such cases, gradients provide more reliable signals to explore this space, especially when there are fewer observations.

**Effect of larger history information.** We relax the Markovian assumption from the earlier experiments and study the impact of including multiple past states and gradient information on generalization to novel tasks. The augmented system is now defined as $h_{\boldsymbol{\varphi}} : \boldsymbol{\theta}_{t-k:t}, \nabla_{t-k:t}, \mathcal{B}_t \to \boldsymbol{\theta}_{t+1}$, where $k$ is a hyperparameter for the number of past states. Our results in Fig. 3 showcase more history information does not impact results much, except in the case of pre-training on FashionMNIST tasks.

**Explicit parameterization is sub-optimal.** Importantly, parametric modeling outperforms the explicit setup highlighting the difficulty of jointly learning $f_{\gamma}$ and $g_{\boldsymbol{\varphi}}$. This is surprising as the class of solutions expressed by the former form a subset of the latter. We attribute this suboptimality to the optimization process as $f_{\gamma}$ and $g_{\boldsymbol{\varphi}}$ are linked in a complicated, non-stationary manner, *i.e.* if the prediction function $f_{\gamma}$ changes then its inference mechanism $g_{\boldsymbol{\varphi}}$ also needs to adapt, and vice versa.

**State parameterization in implicit amortization.** We analyze the design choices associated with the state that persists across iterations to be − (a) Pre-MLP: high-level latent variables tied to the query, (b) Logits: current prediction before undergoing parameter-free normalization like softmax, or (c) Softmax: current predictions after softmax. Fig. 4 shows that modeling the state at logits leads to best performance, and in general shows the importance of being close to prediction for greedy refinement.

**Runtime Efficiency of Amortization.** We note that for a $K$-step iterative method with mini-batch size $B$ leads to a runtime cost of $O(KB^2)$. In contrast, a one-step model runs in $\mathcal{O}(B^2)$ but uses $K$ times less data. When given access to the same amount of data, the one-step model incurs runtime of $\mathcal{O}((KB)^2)$. Thus, iterative amortization is $K$ times more efficient than one-step models when processing the same amount of data. A similar argument applies to memory usage, making the iterative approach more scalable for large tasks. We next compare the amortized parametric model to an off-the-shelf optimizer (Adam) with standard normal initialization as well as MAML (Finn et al., 2017): Adam achieves a loss of 112 while MAML achieves 44.7 in 100 steps, both taking

---

[8]Here, more steps imply steps to infer the vector field $v_t(\cdot|\mathcal{D}_{\mathcal{T}})$ for each $t$, and not more integration steps.

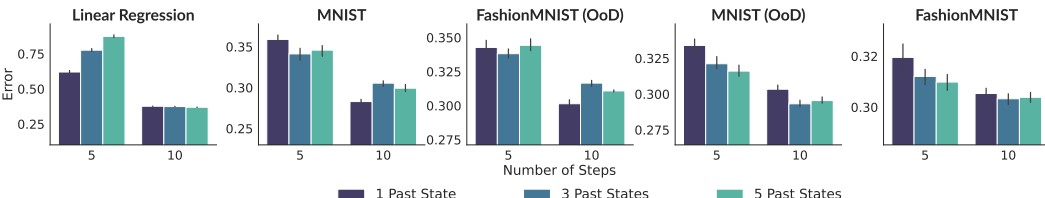

Figure 3: We analyze the benefits, or lack thereof, of leveraging multiple past states and gradients for *parametric amortization*, when we use both observations and gradients as conditional inputs.

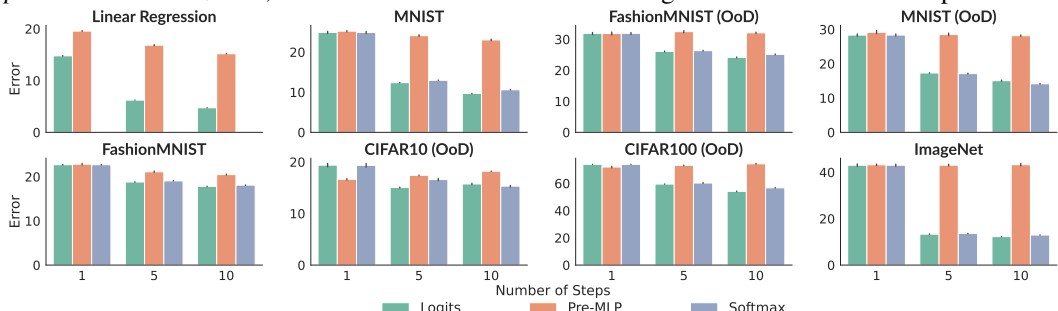

Figure 4: Our ablations reveal that carrying over logits as the recurrent state across iterations in implicit models outperforms other state representations, with softmax outputs performing comparably.

| Method | Parameterization | Input | State / Output | Baselines |
|---|---|---|---|---|
| Parametric | Learned $g_\varphi$ 
 Fixed $f$ | Data and/or 
 Gradients, Query | Parameters | Hypernetworks (1-Step, Data) 
 Learned Optimizers (Multi-Step, Grad) |
| Explicit | Learned $g_\varphi$ 
 Learned $f_\gamma$ | Data and/or 
 Gradients, Query | Latents | Conditional Neural 
 Process (1-Step, Data) |
| Implicit | Fixed $g$ 
 Learned $f_\gamma$ | Data and 
 Query | Prediction | In-Context Learning (1-Step, Data) 
 Prior Fitted Networks (1-Step, Data) |

Table 7: We provide a clear and concise dichotomy of various amortization frameworks along with the baselines considered in each case. The state defines the object which amortizes or adapts to new datasets, which is then used to map to the final prediction for the query.

approximately $0.396$ seconds. In contrast, a single-step amortized parametric model can achieve the performance of $\approx 16$ with inference taking only $0.05$ seconds, thus illustrating the importance of amortized learners which do not rely on off-the-shelf optimizers at inference.

## 6 CONCLUSION

Amortized learning serves as a powerful paradigm for rapid generalization through reuse of computations. By introducing a unified framework and taxonomy, we clarify how different approaches internalize or externalize task adaptation. This perspective not only highlights shared principles but also exposes common bottlenecks: the limited scalability of existing methods in handling large task datasets as well as suboptimality when solely leveraging gradients with fixed functional mappings. To overcome this, we propose iterative amortization, a novel strategy that incrementally refines solutions through mini-batch updates, effectively merging the strengths of optimization-based and forward-pass approaches consistently across parametric, explicit and implicit frameworks for predictive and generative ID and OoD tasks. This framework paves the way for scalably exploring more sophisticated forms of persistent memory across mini-batches for robust task adaptation in increasingly complex, non-iid environments. Additionally, future work involves incorporating richer learning frameworks to optimize Eq. (5) beyond greedy refinement, such as evolutionary algorithms or reinforcement learning. Finally, we believe that such methods can be leveraged for continual and lifelong learning, where the state is continually updated based on non-stationary environment observations.

ETHICS STATEMENT

We focus on the problem of general-purpose amortized learning which is aimed to enhance usefulness of such systems to novel problems solely through inference, but it also carries risks associated with misuse and unintended generalization. Careful evaluation and responsible deployment are essential as these methods become more and more powerful.

REPRODUCIBILITY STATEMENT

We provide all the details to reproduce our results in the appendix.

LLM USE

LLMs were used to assist paper editing.

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

# A RELATED WORK

In this section, we formalize and contrast several existing related frameworks encapsulated in our setting. We highlight the technical differences among these approaches through formal definitions and equations, while retaining key references.

## A.1 META-LEARNING

Meta-learning generally refers to the problem of learning a model (which itself could be a learner, optimizer, or algorithm) that can quickly and efficiently generalize to novel tasks. Formally, let there be a distribution over tasks $p(\mathcal{T})$, with each task $\mathcal{T}$ associated with a dataset $\mathcal{D}_{\mathcal{T}} = \{(\mathbf{x}_j, \mathbf{y}_j)\}_{j=1}^N$. The objective is to learn meta-parameters $\boldsymbol{\theta}$ that enable fast adaptation to new tasks $\mathcal{T}'$ with dataset $\mathcal{D}_{\mathcal{T}'}$:

$$\min_{\theta} \mathbb{E}_{\mathcal{T}} \mathcal{L}\left(\mathcal{D}_{\mathcal{T}}^{\text{valid}}, U_k(\boldsymbol{\theta}, \mathcal{D}_{\mathcal{T}}^{\text{train}})\right), \tag{10}$$

where:

- $U_k(\boldsymbol{\theta}, \mathcal{D}_{\mathcal{T}}^{\text{train}})$ is an adaptation operator applying $k$ gradient steps or another procedure starting from $\boldsymbol{\theta}$,
- $\mathcal{L}\left(\mathcal{D}_{\mathcal{T}}^{\text{valid}}, \cdot\right)$ is the loss on held-out data $\mathcal{D}_{\mathcal{T}}^{\text{valid}}$.

The popular Model-Agnostic Meta-Learning (MAML) framework (Finn et al., 2017) fits into this paradigm with the update:

$$\boldsymbol{\theta}' = \boldsymbol{\theta} - \alpha \nabla_{\boldsymbol{\theta}} \mathcal{L}\left(\mathcal{D}_{\mathcal{T}}^{\text{train}}, \cdot\right) \tag{11}$$

where the meta-objective optimizes $\boldsymbol{\theta}$ such that $\boldsymbol{\theta}'$ performs well on $\mathcal{D}_{\mathcal{T}}^{\text{valid}}$. Extensions such as Reptile (Nichol & Schulman, 2018) and Meta-SGD (Antoniou et al., 2018) also follow this principle. These methods ultimately fit within the *parametric modeling* framework, as the model parameters $\boldsymbol{\theta}$ are explicitly adapted to new tasks (Raghu et al., 2019). Here, the adaptation operator $U_k$ can be seen as the analogue of $g_{\boldsymbol{\varphi}}$ from our setting.

## A.2 AMORTIZATION

Amortized inference traditionally refers to learning an inference network $q_\phi(\mathbf{z}|\mathbf{x})$ to approximate the posterior $p(\mathbf{z}|\mathbf{x})$, as in variational autoencoders (Kingma et al., 2013; Rezende et al., 2014). Our focus differs by considering amortization at the dataset or task level, rather than per-observation. We formalize this as learning an amortized posterior:

$$q_\phi(\mathbf{z} \mid \mathcal{D}) \approx p(\mathbf{z} \mid \mathcal{D}), \tag{12}$$

where $\mathcal{D}$ is a dataset corresponding to a task. This approach implicitly or explicitly learns an optimizer or inference model that can solve novel tasks *zero-shot*. This viewpoint aligns with probabilistic meta-learning frameworks (Garnelo et al., 2018b; Amos et al., 2023), which emphasize amortizing inference across a distribution of tasks. Such methods can be framed as parametric approaches when the target $\mathbf{z}$ of interest are all the parameters of the likelihood (Mittal et al., 2025b) or explicit approaches when the likelihood is learned as well and $\mathbf{z}$ just represents some latents (Garnelo et al., 2018a;b; Requeima et al., 2019).

In more general terms, all methods of meta-learning rely on some or the other notion of amortization and depending on the object that is trained to be amortized, we recover parametric, explicit or implicit models.

## A.3 LEARNED OPTIMIZERS

Learned optimizers learn parameter update functions $h_{\boldsymbol{\varphi}}$ conditioned on gradients and optimization states:

$$\boldsymbol{\theta}_{t+1} = \boldsymbol{\theta}_t + h_{\boldsymbol{\varphi}}(\nabla_{\boldsymbol{\theta}} \mathcal{L}(\mathcal{B}_{\mathcal{T}}, \boldsymbol{\theta}), \mathbf{s}_t), \tag{13}$$

where $\mathbf{s}_t$ denotes the optimizer's internal state, e.g., momentum or recurrent memory and $\mathcal{B}_\mathcal{T}$ represents a mini-batch of task-specific training data. These methods have been studied extensively (Metz et al., 2022b; Knyazev et al., 2024; Metz et al., 2019; 2022a; Li & Malik, 2017; Wichrowska et al., 2017).

While learned optimizers enable scalable amortization of inference, their hypothesis class is limited by dependence on *only first-order gradient information* and parameter updates. They cannot easily represent complex second-order dynamics such as Newton methods, which rely on Hessian information. Furthermore, they require explicit task parameters $\boldsymbol{\theta}$, restricting them to parametric models and preventing direct applicability to implicit modeling approaches. Here, one can see the recursive application of $h_\varphi$ as the $g_\varphi$.

### A.4 HYPERNETWORKS

Hypernetworks (Ha et al., 2016) are architectures that take a dataset $\mathcal{D}$ as input and generate the weights $\boldsymbol{\theta}$ of a target neural network to be used for predictions on that dataset:

$$\boldsymbol{\theta} = H_\varphi(\mathcal{D}) \tag{14}$$

with the target model output

$$\mathbf{y} = f(\mathbf{x}, \boldsymbol{\theta}) = f(\mathbf{x}, H_\varphi(\mathcal{D})) \tag{15}$$

Such models have been studied extensively (Krueger et al., 2017; Chauhan et al., 2024; Schug et al., 2024; Von Oswald et al., 2019; Bronskill et al., 2021), including recent work that shows standard transformers can be interpreted as hypernetworks (Schug et al., 2024). Hypernetworks are generally *parametric* since $\boldsymbol{\theta}$ is explicitly generated. However, if the hypernetwork conditions directly on the query $\mathbf{x}$, it can realize implicit models within our taxonomy. The general case of hypernetworks can, however, be seen as parametric modeling with $H_\varphi$ being the $g_\varphi$ component.

### A.5 IN-CONTEXT LEARNING AND PRIOR-FITTED NETWORKS

In-context learning (ICL) solves similar problems by training models (notably transformers) to make predictions conditioned on example input-output pairs provided as context, either by modeling explicit parameters akin to hypernetworks (Mittal et al., 2025b;a) or without explicit parameter updates (Mittal et al., 2024; Elmoznino et al., 2024; Müller et al., 2021; Garg et al., 2022; Von Oswald et al., 2023; Hollmann et al., 2022; Li et al., 2023; Bai et al., 2023). In addition, (Wu et al., 2025; Wang et al., 2024; Bai et al., 2023) highlight transformer-based implicit ICL can actually express a large family of meta-learners, ranging from gradient-based methods to neural processes.

Formally, given a context $\mathcal{C} = \{(\mathbf{x}_i, \mathbf{y}_i)\}_{i=1}^K$, the model learns to produce predictions $f_\gamma(\mathbf{x}, \mathcal{D})$ with no parameter updates performed during inference. ICL and PFNs share the same modeling setup when the input is a sequence of examples, though ICL can be more general by conditioning on other forms of task information such as language prompts.

Both are subsumed within our unified framework, categorized as *parametric*, *explicit*, or *implicit* models depending on how the in-context learner is parameterized.

## B  OVERALL PIPELINE

We additionally provide a rough sketch that walks through our pipeline, which consists of two distinct phases.

### B.1 META-TRAINING

For all methods — parametric, explicit, or implicit — we train an amortized model on a diverse set of tasks. All frameworks train a Transformer as the amortized network, with an optional MLP as the predictor $f_\gamma$ for the explicit setup. Each task is either derived from a dataset (*e.g.* , MNIST or ImageNet) or synthetically generated (e.g., linear regression tasks). Details of task construction are in Section 5 and Appendix C, but we provide a concrete example here:

For MNIST, each task involves applying:

- A random projection from the original $28 \times 28$ image space to a 100-dimensional space.
- A random permutation of class labels (e.g., all images labeled '5' may be relabeled as '2' and so on).

**Training Procedure.** All methods are trained using maximum likelihood: the objective is to predict a query's label given a batch of context examples and the model's current state. However, training how the state updates is only done greedily – given a state and mini-batch, the model is trained to predict the next best state without taking into account how it impacts future states.

**Distinction Between Method Variants.** Owing to the unified framework, we can now succinctly describe the key difference across different amortization methods simply as how this state is defined:

- Parametric: The state is the weights of a linear predictor, fixing the mapping from state and query to prediction (denoted $f$).
- Explicit: The state is a learned high-dimensional latent vector. It is updated using context examples and gradients, and is then used by a learned prediction function (denoted $f_\gamma$).
- Implicit: The state is defined per-query, representing the model's belief about the correct label for that specific query. Note that in this case the state is not a function of the dataset, but a function of (dataset, query).

## B.2 EVALUATION

**In-Distribution** After meta-training, we evaluate the model on in-distribution tasks, i.e., tasks drawn from the same distribution as the training tasks. For example, with MNIST:

- A new random projection matrix is sampled (from the same distribution used during training).
- A subset of training examples are projected to form the context set.
- The model then makes predictions on the corresponding projection of test examples.

This setup evaluates whether the amortized model generalizes to unseen tasks that are structurally similar to those seen during training (i.e. new random projections and new test images).

**Out-of-Distribution (OoD).** To assess cross-dataset generalization, we also evaluate the model on out-of-distribution tasks. For example:

- We evaluate the MNIST model on FashionMNIST — context and queries are drawn from the train and test set of FashionMNIST respectively, following the same procedure for random projection as in-distribution.

This tests whether the learned algorithm transfers to tasks with different input distributions. Note that for ImageNet experiments we do not use random projections but instead rely on Dino-v2 embeddings.

## C DATASET DETAILS

In this section, we provide details of all the datasets and tasks considered for pre-training the different variants of amortized estimators as well as the tasks used for evaluation.

## C.1 LINEAR REGRESSION

For the problem of linear regression, we define each task with a corresponding weight vector $\mathbf{w}_\mathcal{T}$ such that observations in the context and query set follow the mapping $(\mathbf{x}, \mathbf{y} = \mathbf{w}_\mathcal{T}^T \mathbf{x} + \epsilon)$ where $\mathbf{x}$ is sampled from a standard normal distribution and $\epsilon$ denotes random noise sampled from a normal distribution with standard deviation of $0.25$. For this problem, we define the family of tasks for pre-training and evaluation corresponding to $\mathbf{w}_\mathcal{T}$ sampled randomly from the unit normal.

For training we pass a set of $\{(\mathbf{x}_i, \mathbf{y}_i)\}_i$ as the context and a set of $\mathbf{x}$ as the query, where the underlying mapping between them is shared to be the same $\mathbf{w}_\mathcal{T}$. The task of the in-context learner is to predict either the coefficients $\boldsymbol{\theta}_\mathcal{T}$ that are fed to a prediction network or known mapping (*explicit* or *parametric*), or it predicts the prediction corresponding to $\mathbf{x}$ directly. Thus, the in-context learner has to learn to model and minimize the underlying validation loss corresponding to the problem.

## C.2 MNIST / FASHIONMNIST CLASSIFICATION

Similar to linear regression, we randomly sample a projection matrix $W_\mathcal{T} \in \mathbb{R}^{784 \times 100}$ from unit normal and a class re-labeling matrix $\pi_\mathcal{T}$ such that $\pi_{\mathcal{T},i,j} \in \{0,1\}$ and $\sum \pi_{\mathcal{T},i,:} = 1$ such that each task is defined with the following transformations on observations and labels:

$$\mathbf{x} \to W_\mathcal{T}\mathbf{x} \quad \mathbf{y} \to \pi_\mathcal{T}\mathbf{y} \tag{16}$$

where $\mathbf{x} \in \mathbb{R}^{784}$ and $\mathbf{y}$ denotes the one-hot vector corresponding to the class. This operation defined above can be seen as randomly projecting pixel values to a lower dimensional space and re-labeling the classification problem such that the same classes are mapped to the same new class − for *e.g.* the remapping could map digits $1, 4, 7$ to class $1$, and so on. That is, this re-labeling can club multiple classes into a single class or rename class labels, but never sends two objects from the same class to different labels.

We follow the same setup for FashionMNIST as well, and then check for OoD evaluation on the other dataset by feeding a set of observations as context and evaluating on queries; note that task specific variables $W_\mathcal{T}, \pi_\mathcal{T}$ are shared between context and query.

## C.3 IMAGENET CLASSIFICATION

We follow the same setup as MNIST / FashionMNIST classification with just a few differences − dimensionality reduction is done using a pre-trained Dino-v2 model and thus this projection operator is shared across tasks and not task-dependent anymore, and each task is defined by sampling a subset of images corresponding to 100 random classes of ImageNet from which grouping / re-labeling is done exactly as above. That is, for any task the problem is to solve the maximum 100-way classification problem as opposed to the single 1000-way problem of ImageNet. Correspondingly, evaluation is done on contexts from ImageNet training set and queries from evaluation set corresponding to multiple 100-way problems as well as CIFAR-10 and CIFAR-100's train and test set forming the context and query.

## C.4 TOPOLOGICAL ORDER PREDICTION

**Problem Statement.** We begin by defining *Structural Causal Models (SCMs)*, which formalize the causal generative process over a set of random variables. An SCM defines a distribution over of $d$ endogenous variables $\boldsymbol{V} = \{X_1, \ldots, X_d\} \sim \mathbb{P}_X$, each determined by a deterministic function of its parents ($F_i$) and a corresponding exogenous noise term ($N_i$). Specifically, each variable $X_i$ is generated as:

$$X_i = F_i(\mathrm{Pa}(X_i), N_i), \quad \text{with} \quad \mathrm{Pa}(X_i) \subseteq \boldsymbol{V} \setminus \{X_i\} \tag{17}$$

where the functions $F_i$ define the causal mechanisms, $\mathrm{Pa}(X_i)$ denotes the set of direct causes (parents) of $X_i$, and $N_i$ is a latent noise variable drawn independently from a distribution $\mathbb{P}_N$. Collectively, the SCM is represented as $\mathcal{S}(\mathbb{P}_N, F, \mathcal{G})$, where $\mathcal{G} \in \{0,1\}^{d \times d}$ is the adjacency matrix of the causal graph, i.e., $\mathcal{G}_{ij} = 1$ if $X_j \in \mathrm{Pa}(X_i)$.

Following standard assumptions in the literature, we consider markovian SCMS, where we restrict the causal graph $\mathcal{G}$ to be a directed acyclic graph (DAG) and the set of noise variables $\{N_i, \cdots, N_d\}$ to be mutually independent.

Given the DAG assumption, we can obtain a unique topological order $\tau$ associated with the causal graph $\mathcal{G}$, and the prediction task is defined as follows.

> Given a dataset of causal variables $D_X \in \mathbb{R}^{n \times d}$ samples from an unknown SCM $\mathcal{S}(\mathbb{P}_N, F, \mathcal{G})$, the goal is predict the associated topological order $\tau$ from $D_X$.

**Method.** For amortized inference of topological order, we follow the approach from Scetbon et al. (2024). They leverage transformer-based architecture that attend over both the sample dimension ($n$) and the node dimension ($d$) to attend over the context, followed by a linear prediction layer to classify the leaf nodes (no outgoing edge) of the causal graph $\mathcal{G}$. Once we obtain the current leaf nodes, we can remove them from the dataset $D_X$ and repeat the procedure again to predict leaf nodes. This recursive procedure will terminate in at most $d$ iteration and output the inferred topological order $\hat{\tau}$.

**Synthetic Data Simulator.** Following (Scetbon et al., 2024), we use the AVICI synthetic data simulator Lorch et al. (2022). This synthetic generator supports diverse structural and functional variations, making it particularly well-suited for training and evaluating models under distribution shifts. We describe below the options avaiable for each compoenent in the SCM.

- *Graph Structures.* We can sample causal graphs from a variety of schemes; Erdős–Rényi graphs (Erdos & Renyi, 1959), Scale-free networks (Barabási & Albert, 1999), Watts–Strogatz small-world graphs (Watts & Strogatz, 1998), and Stochastic block models (Holland et al., 1983).
- *Noise Distributions.* Exogenous noise variables $\{N_i\}$ are sampled from either Gaussian or Laplace distributions, with randomly selected variances.
- *Functional Relationships.* Causal relationships are instantiated using either Linear (LIN) models with randomly sampled weights and biases, or with Random Fourier Features (RFF) to generate more complex, non-linear mappings.

To simulate distribution shifts, two families of SCM distributions are defined:

- *In-Distribution* ($\mathbb{P}_{\text{in}}$): Graphs are sampled from Erdős–Rényi and scale-free models, noise from Gaussian distributions, and functions are either LIN or RFF using standard parameter ranges.
- *Out-of-Distribution* ($\mathbb{P}_{\text{out}}$): Graphs are drawn from Watts–Strogatz or stochastic block models, noise from Laplace distributions, and functions (LIN/RFF) are sampled from disjoint parameter ranges to introduce a shift.

*Parameter Ranges:*

- Linear Mechanisms:
  - $\mathbb{P}_{\text{in}}$: weights $\sim U_{\pm}(1, 3)$, bias $\sim U(-3, 3)$
  - $\mathbb{P}_{\text{out}}$: weights $\sim U_{\pm}(0.5, 2) \cup U_{\pm}(2, 4)$, same bias range
- RFF Mechanisms:
  - $\mathbb{P}_{\text{in}}$: length scale $\sim U(7, 10)$, output scale $\sim U(5, 8) \cup U(8, 12)$, bias $\sim U_{\pm}(-3, 3)$
  - $\mathbb{P}_{\text{out}}$: length scale $\sim U(10, 20)$, output scale $\sim U(8, 12) \cup U(18, 22)$, bias $\sim U_{\pm}(-3, 3)$

*Train Datasets.* We train the model by randomly sampling SCMs from the $\mathbb{P}_{\text{in}}$ distribution. Each epoch contains datasets with $n = 1000$ samples and $d = 20$ nodes.

*Test Datasets.* We evaluate performance across four distinct settings that differ in the distribution they induce over SCMs. From each SCM, we sample a two test datasets, one with $n = 1000$ samples and $d = 20$ nodes; and the with $n = 1000$ samples and $d = 50$ nodes.

- `LIN`: Linear mechanisms under $\mathbb{P}_{\text{in}}$; total of 9 randomly sampled SCMs with 3 different graph types.
- `RIN`: Nonlinear (RFF) mechanisms under $\mathbb{P}_{\text{in}}$; total of 9 randomly sampled SCMs with 3 different graph types.
- `LOUT`: Linear mechanisms under $\mathbb{P}_{\text{out}}$; total of 6 randomly sampled SCMs with 2 different graph types.
- `ROUT`: Nonlinear (RFF) mechanisms under $\mathbb{P}_{\text{out}}$; total of 6 randomly sampled SCMs with 2 different graph types.

### C.5 ALPHABETS

We next turn our attention to generation tasks where the goal is to generate novel samples by inferring the underlying density defined by the context examples and consequently learning to sample from it. We first borrow the task of sampling point clouds resembling alphabets (Atanackovic et al., 2024) where different tasks $\mathcal{T}$ correspond to different alphabets with different scaling and rotation operations applied. Out of all the capitalized alphabets in the English language, we reserve {D,H,N,T,X,Y} solely for evaluation, *i.e.* they are not provided for training on during pre-training. We then consider evaluation on both in-distribution and out-of-distribution alphabets.

### C.6 Gaussian Mixture Model

Similar to the above, we consider the problem of sampling similar to a mixture of Gaussians provided as in-context examples. Here, each task is defined by an underlying number of clusters $n_{\mathcal{T}}$ as well as corresponding means $\mu_{\mathcal{T},i}$ which are sampled from a normal distribution with 5 standard deviation. Samples from this GMM are obtained by fixing the standard deviation corresponding to each cluster as 0.3 and then drawing samples from this mixture distribution. We train the in-context estimator on a constant stream of new tasks synthetically generated and then evaluate them on new tasks sampled randomly. Our tasks involve both a simple version of GMM which is in 2-dimensional observed space as well as a more complex GMM which is 5-dimensional. The number of clusters are uniformly sampled from a maximum of 100 clusters.

## D Metrics

For metrics, we consider the $l_2$ loss for the regression problem while for the classification problems, we consider error as a metric, which is $100-$Accuracy, which we use for the topological prediction task as well. For the generative modeling tasks, we consider the 2-Wasserstein and 1-Wasserstein distance which are based on optimal transport and can be described as

$$\mathcal{W}_p = (\inf_{\pi} \frac{1}{N} \sum_i \|\mathbf{x}_i - \hat{\mathbf{x}}_{\pi_i}\|_p^p)^{1/p} \tag{18}$$

where $\pi$ describes a permutation matrix, of which we use $p = 1$ or $p = 2$.

## E Experimental Details

### E.1 Model Parameterization

We parameterize the amortized model, $g_{\boldsymbol{\varphi}}$ in the case of parametric and explicit model, and $f_{\gamma}$ in the case of implicit model, as a transformer with 512 hidden dimensions, 2048 hidden dimensions in the feed-forward neural network and 8 heads and 8 layers. We use gelu activation function and perform normalization first in PyTorch's version of transformers. In addition, we consider a specific learnable linear encoder for gradients as well as observations, where observations are $(\mathbf{x}, \mathbf{y})$ pairs and gradients correspond to $\overline{\nabla}_{\boldsymbol{\theta}}\mathcal{L}(\mathbf{y}, f_{\gamma}(\mathbf{x}, \boldsymbol{\theta}))|_{\boldsymbol{\theta}_{\mathcal{T}}}$ for parametric and explicit models while implicit models cannot use gradients.

In addition, for implicit models we append $\hat{\mathbf{y}}$ to query observations and re-use the observation encoder to embed queries in the same space. We additionally use a learnable query embedding that is added to the query tokens for implicit model to be able to differentiate context from query. For the ablation conducted on Pre-MLP state that is carried over, we leverage a separate learnable query embedding matrix that does not share weights with the observation encoder. Finally, to model predictions or vector fields in the case of generative modeling, we leverage a similar linear decoder that maps from 512 dimensions to dimensions of the prediction, parameters, latents or vector field depending on the task and modeling setup.

It is important to note that our model should be able to make predictions or model parameters conditioned on arbitrary number of context examples, and hence should be trained in a manner that allows for varying context length $-$ this is similar to decoder only transformers that learn to make predictions for every context length in parallel. Given the difference in our setting from autoregressive language modeling, we provide two ways of implementing this amortization through the use of either a causally masked transformer or a non causal transformer, the details of which are provided in the next subsections.

Finally, for modeling iterative amortized inference, we use a simple setup where the output from one batch $-$ $\boldsymbol{\theta}$ or $\hat{\mathbf{y}}$ depending on the amortization framework used $-$ is fed back into the transformer with another batch after being detached from the computation graph. Here, $\boldsymbol{\theta}$ is just fed as another token with or without its gradient information as an additional token, while for implicit models $\hat{\mathbf{y}}$ is appended to its corresponding query, hence the state represents the predictions corresponding to the query that get updated after every iteration $-$ note that the state is tied to a query here. Given a number of steps, $e.g.$ $k$, we iterate over this process $k$ times and compute gradients each time,

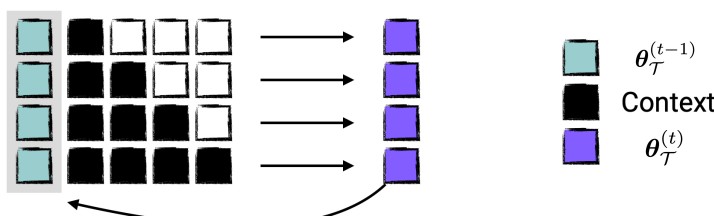

Figure 5: Masking procedure for the causally masked parametric and explicit model, where the context evolves according to a causal mask − the matrix of black and white squares describes the masking procedure where white blocks denote masks in the matrix − where each token in parallel consequently predicts parameters of interest conditioned on past batch of data and previous state. This mimics having variable sized dataset and processing for variable dataset sizes in parallel. The output corresponding to the last token is the $\boldsymbol{\theta}_{\mathcal{T}}^{(t)}$ that gets fed back recurrently.

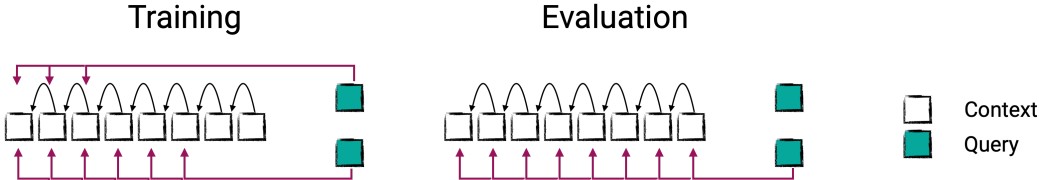

Figure 6: Masking procedure for the causally masked implicit model, where the context evolves according to a causal mask and the queries can attend somewhere in between, which conditions them on that particular position and the positions behind it. Since we use multiple queries in parallel which can look at different number of past contexts through this manipulation of the masking, we obtain parallel processing of multiple dataset sizes.

accumulating them before taking a gradient step. Given that we detach the state before feeding it again, it prevents backpropagation through time and thus defines a greedy procedure.

### E.2 TRAINING OBJECTIVE

The training procedure for all the models can be seen as optimization over the following loss for parametric and explicit models, where the batch sizes can contain variable number of observations $N$ which is randomly sampled from $1 - 100$.

$$\arg\min_{\gamma, \boldsymbol{\varphi}} \frac{1}{T} \sum_{t=1}^{T} \mathbb{E}_{\mathcal{T}} \mathbb{E}_{\mathcal{B}_{\mathcal{T}}^{\text{valid}} \subseteq \mathcal{D}_{\mathcal{T}}^{\text{valid}}} \mathbb{E}_{\mathcal{B}_{\mathcal{T}}^{(t),\text{train}} \subseteq \mathcal{D}_{\mathcal{T}}^{\text{train}}} \frac{1}{N} \sum_{(\mathbf{x},\mathbf{y}) \in \mathcal{B}_{\mathcal{T}}^{\text{valid}}} \mathcal{L}(\mathbf{y}, f_\gamma(\mathbf{x}, g_{\boldsymbol{\varphi}}(\mathcal{B}_{\mathcal{T}}^{(t),\text{train}}, \boldsymbol{\theta}_{\mathcal{T}}^{(t)}))) \quad (19)$$

$$\boldsymbol{\theta}_{\mathcal{T}}^{(t)} = g_{\boldsymbol{\varphi}}(\mathcal{B}_{\mathcal{T}}^{(t-1),\text{train}}, \text{sg}(\boldsymbol{\theta}_{\mathcal{T}}^{(t-1)})) \quad (20)$$

In contrast, implicit models consider the following learning paradigm

$$\arg\min_{\gamma, \boldsymbol{\varphi}} \frac{1}{T} \sum_{t=1}^{T} \mathbb{E}_{\mathcal{T}} \mathbb{E}_{\mathcal{B}_{\mathcal{T}}^{\text{valid}} \subseteq \mathcal{D}_{\mathcal{T}}^{\text{valid}}} \mathbb{E}_{\mathcal{B}_{\mathcal{T}}^{(t),\text{train}} \subseteq \mathcal{D}_{\mathcal{T}}^{\text{train}}} \frac{1}{N} \sum_{(\mathbf{x},\mathbf{y}) \in \mathcal{B}_{\mathcal{T}}^{\text{valid}}} \mathcal{L}(\mathbf{y}, f_\gamma(\mathbf{x}, \mathcal{B}_{\mathcal{T}}^{(t),\text{train}}, \hat{\mathbf{y}}^{(t)}))) \quad (21)$$

$$\hat{\mathbf{y}}^{(t)} = f_\gamma(\mathcal{B}_{\mathcal{T}}^{(t-1),\text{train}}, [\mathbf{x}, \text{sg}(\hat{\mathbf{y}}^{(t-1)})]) \quad (22)$$

### E.3 CAUSAL MASKED MODEL

Given that the implicit model differs considerably from the parametric and explicit model in terms of conditional independence assumptions defined, we need to handle the two cases somewhat differently to enable efficient parallelized implementations. We discuss the details below.

**Parametric and Explicit**. For the parametric and explicit model, we consider a simple causal masking over observations such that the model predicts $\boldsymbol{\theta}_{\mathcal{T}}$ corresponding to each token conditioned

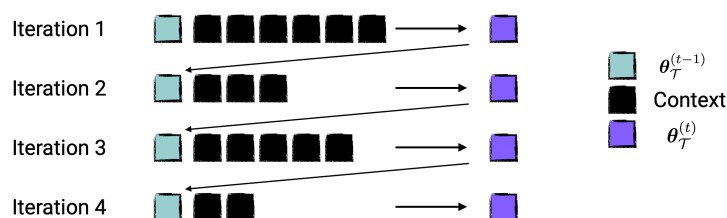

Figure 7: We show the case of non causal masked transformer for parametric and explicit setup where at each iteration, a differently sized context is provided as input to predict the next state. The size of the context is randomly sampled. At evaluation we use the full context.

on the previous token, and the loss is aggregated by using all the intermediate $\boldsymbol{\theta}_{\mathcal{T}}$'s obtained to make predictions on a validation set and averaging them. This procedure is highlighted in Fig. 5.

**Implicit**. In contrast to the parametric approach, the implicit framework cannot deal with this in such a simple manner since the query is $\mathbf{x}$ which cannot remain in the same position as well as attend over variable sized context. To resolve this issue, we process in parallel multiple points from the validation set and manipulate the masking matrix to randomly let them attend to some token $j$ and its predecessors. We use a causal transformer so this ensures that if the query looks at index $j$ and before, then that query models prediction using a dataset of size $j$ instead of context length. We use multiple queries in parallel, randomly choosing the corresponding $j$ so that we can, in-parallel, model predictions for variable sized datasets in a single forward pass (though they have to correspond to different queries). A pictorial representation of this is showcased in Fig. 6.

### E.4 Non-Causal Model

For the non causal masked version, at each training iteration we just randomly sample a length $n$ and then consider a batch of only $n$ observations for a forward pass. The benefits of this approach is that it can allow for conditioning on the $n$ observations in-context without having to rely on a causal decomposition of attention but the downside is that the number of observations have to be kept fixed for a training iteration leading to unbiased gradients but they may have high variance since each training iteration only relies on a single sample ($n$) of the number of observations which then differ across training iterations. Thus, it is not possible to leverage parallel computation for a variety of number of observations within a single forward pass.

A benefit of this approach is that for parametric and explicit models, it can allow more than one token to denote the latents which is computationally infeasible for the masked case. For explicit models in particular, we consider 8 tokens for latents which are of 128 dimensions each; note that having this for masked case is possible but it will require complex masking procedures as well as efficient sparse attention kernels otherwise the context length would blow up relatively quickly.

We refer to Figs. 7 and 8 for details regarding the non causal transformer model for the parametric, explicit and implicit cases.

## F    Additional Ablations

### F.1    Causal vs Non Causal Transformer

We perform a thorough comparison between using a causal and non-causal transformer in all three amortization regimes − *parametric*, *explicit*, and *implicit*. We generally see mixed results which we hypothesize is because while the non-causal method is more expressive algorithm, it has a higher variance of gradients during training as one cannot process in parallel multiple dataset sizes and thus leverage larger effective batch size which makes them inefficient to train and the optimization process largely more unstable.

**Parametric**.

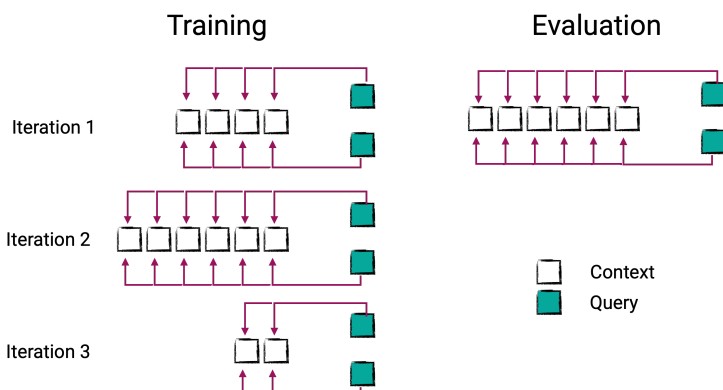

Figure 8: We highlight the case of non-causal transformer model for the implicit model where at each iteration a subset of observations are provided as context; their number randomly sampled. At evaluation we use the full context.

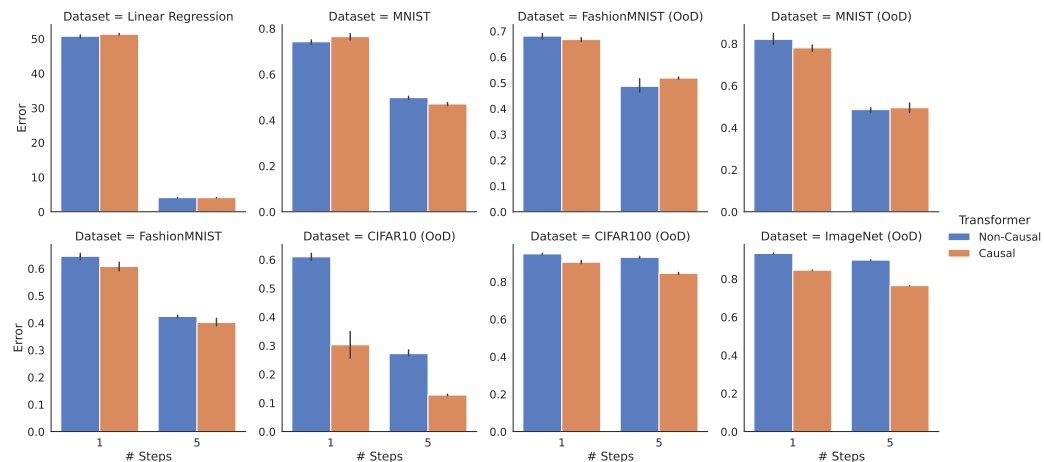

Figure 9: **Parametric with Gradient Information**. We look at the comparison between causal and non-causal transformer for a parametric model that leverages gradients but not data.

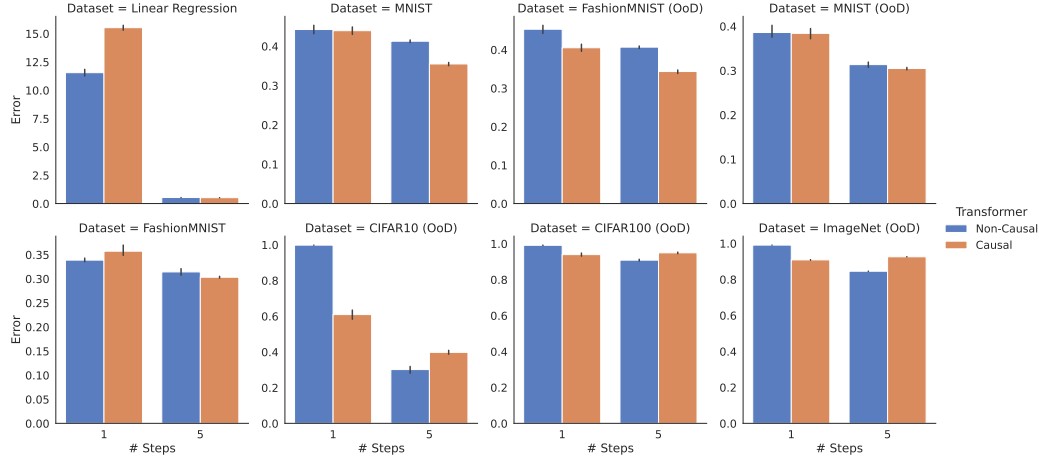

Figure 10: **Parametric with Observations Information**. We look at the comparison between causal and non-causal transformer for a parametric model that leverages data but not gradients.

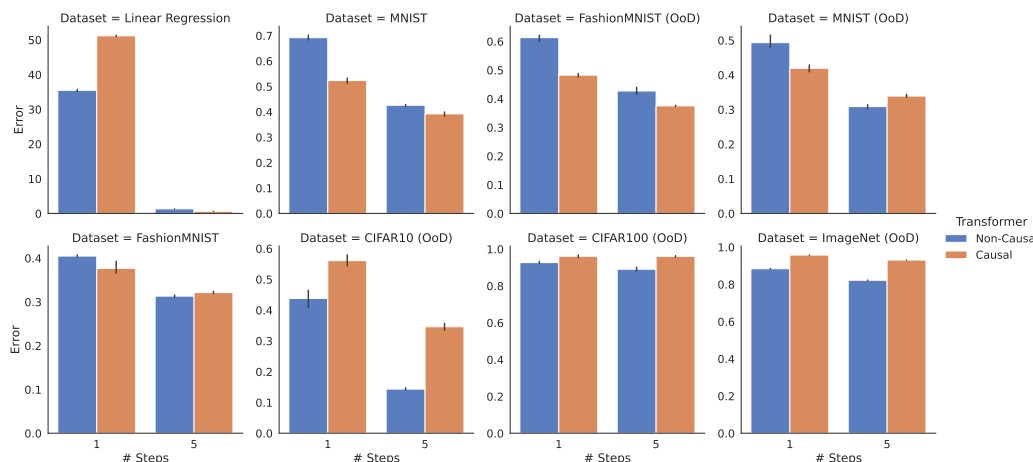

Figure 11: **Parametric with Observations and Gradient Information**. We look at the comparison between causal and non-causal transformer for a parametric model that leverages both gradients and data.

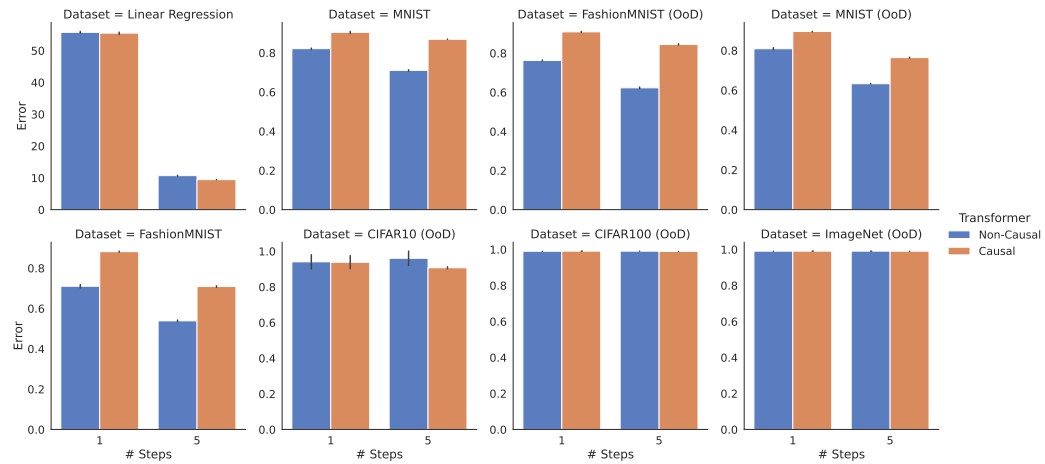

Figure 12: **Explicit with Gradient Information**. We look at the comparison between causal and non-causal transformer for an explicit model that leverages gradients but not data.

For the parametric model, we consider the causal method that predicts weights of the linear predictor in parallel conditioned on $\mathbf{x}_{1:n}$ for all $n$ in parallel by leveraging a causal mask, as described in the section above whereas the non-causal framework leverages a more expressive transformer since the attention is not restricted to causal attention but at every training iteration, a single $n$ is sampled at random and conditioning is done on only a subset of $n$ observations (this $n$ is changed at every iteration). We can see that this is an unbiased estimate of the gradient but with only a single sample of $n$, and thus suffers from larger variance but allows for more expressive architecture. We highlight the performance difference of the two approaches for various number of iterations of amortization as well as modalities of information − gradient or observations or both − in Figs. 9 to 11.

**Explicit**.

Similar to the parametric modeling setup, we look at the explicit amortization in the case of both a causal and a non-causal transformer with different number of iterations as well as different modalities of information − gradient or observations or both − in Figs. 12 to 14.

**Implicit**. We refer to Fig. 15 for experimental details contrasting the causal and non-causal transformer model.

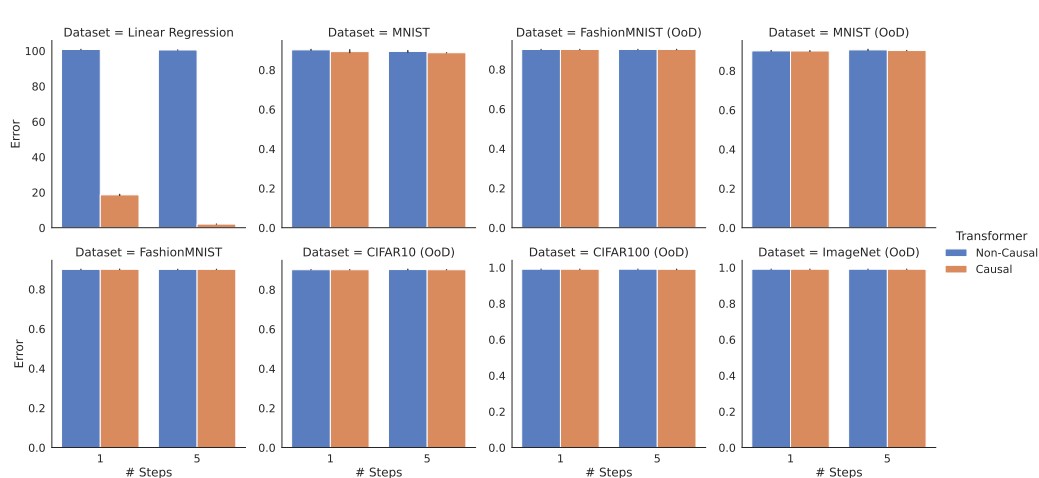

Figure 13: **Explicit with Observation Information**. We look at the comparison between causal and non-causal transformer for an explicit model that leverages data but not gradients.

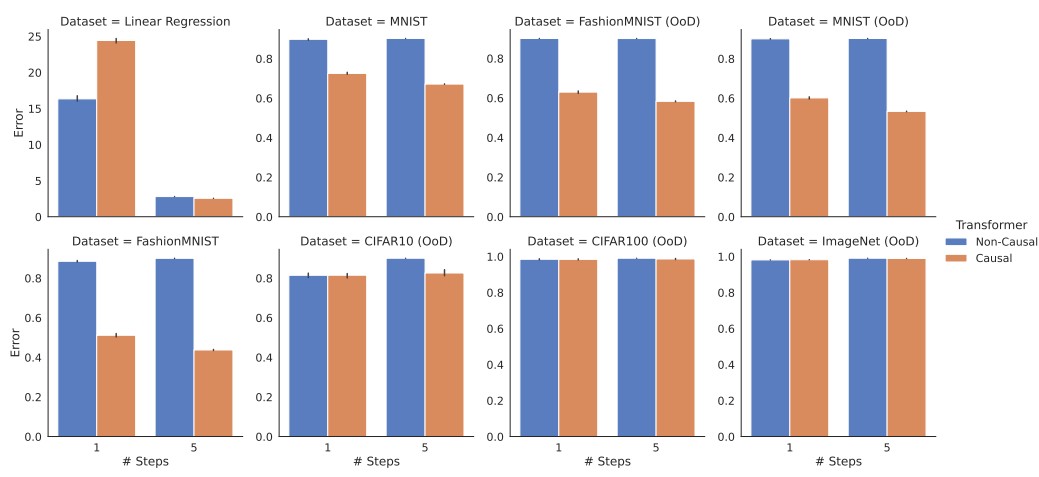

Figure 14: **Explicit with Observation and Gradient Information**. We look at the comparison between causal and non-causal transformer for an explicit model that leverages both gradients and data.

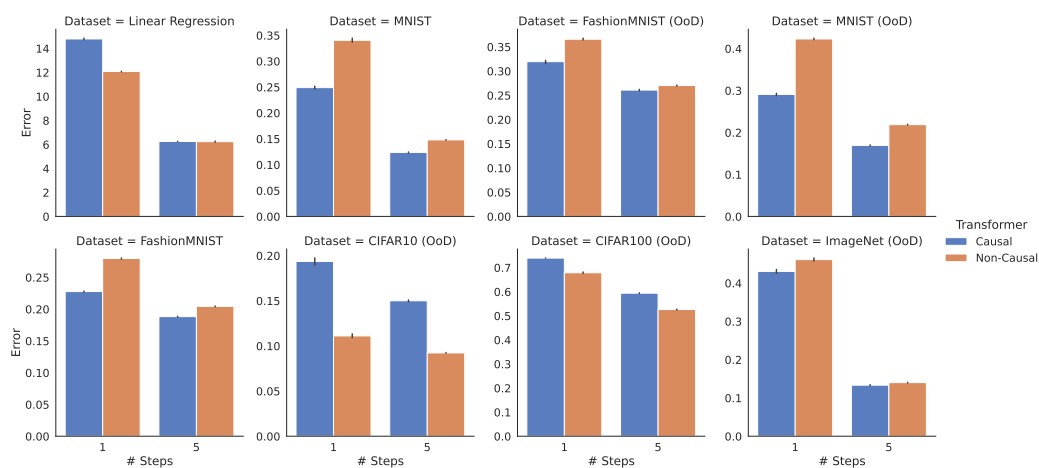

Figure 15: **Implicit with Observation Information**. We look at the comparison between causal and non-causal transformer for an implicit model, which can only leverage data.

As highlighted at the start of the section, our results on the comparisons are mixed as one model is more expressive while the other leads to optimization challenges due to potential variance in gradients since it is a single sample monte carlo estimate.

