# OpenReview forum: "Iterative Amortized Inference: Unifying In-Context Learning and Learned Optimizers"
_ICLR.cc/2026/Conference — Submitted to ICLR 2026_

### Official Review · Reviewer_LiiH · 2025-10-21

**Soundness:** 3
**Presentation:** 2
**Contribution:** 4
**Rating:** 8
**Confidence:** 3

**Summary:**

This paper unifies meta-learning, learned optimizers, in-context learning, etc. into a common framework of amortized learning, i.e. learning how to reuse shared information across tasks to improve performance. In this framework, methods vary along two axes, how meta-testing data is converted into task-specific parameters, and how those parameters are combined with queries to obtain predictions. Those two processes are either fixed or learnable, leading to three categories of methods. Based on the categorization, the paper then proposes two general methods that iteratively refines task-specific parameters using mini-batches of meta-testing data (which subsumes some existing algorithms). Experiments are done on regression, classification, and generation datasets showing the power of the framework.

**Strengths:**

This paper provides a framework that unifies three disparate ideas, meta-learning, learned optimizers, and in-context learning. The connection of in-context learning to the others is especially insightful. I believe that this kind of work that combines different topics and provides a bird's-eye view is valuable for the community, as it has the potential to bring to light new avenues of research.

The framework itself is mathematically rigorous and clearly defined, and it encompasses many previous algorithms. Experiments are done on a variety of tasks, including classification, regression, and generation. Results provide practical lessons, e.g. parametric models outperform explicit models.

**Weaknesses:**

The experiments are incomplete in some respects. First, only one network architecture is considered - it would strengthen the work to see how the method scales. Second, the parametric and explicit models only consider regression and classification tasks, but they can also be used for generation.

Moreover, the paper can be unclear at times. The main idea is abstract, so more explanation would be helpful, for example by:
- Adding an explanation to the caption of figure 1
- Explaining what likelihood is being referred to in line 206
- In line 322, explaining what is the causal structure being masked
- At the end of section 4, comparing and contrasting the proposed methods to existing algorithms

**Questions:**

The errors in table 2 are quite large. Is there any intuition why?

---

> ### Author Response · Authors · 2025-11-18
>
> We thank the reviewer for their useful feedback and have updated the manuscript with their suggestions. We hope that our response addresses the concerns raised by the reviewer.
>
> **Network architectures**: We consider the standard transformer for all our experiments given its strong performance across various modalities (language, vision, amortized prediction). [3] explores the contrast between different architectures for amortized Bayesian posterior estimation and highlights that Transformers remain the most performant architecture among counterparts (RNNs or DeepSets).
>
> We do analyze the trade-off between a causal versus non-causal transformer and find that the causal variant performs better in practice. Although the non-causal model is strictly more expressive, it is significantly less stable during training. The key advantage of the causal transformer is that it enables conditioning on multiple context lengths within a single forward pass, effectively increasing the batch size through parallelization and thereby improving stability and efficiency. This is not feasible with a non-causal architecture. Experimental comparisons between the two can be found in Appendix F.
>
> **Parametric / Explicit amortization in generative models**: We completely agree that parametric and explicit frameworks can theoretically be used in generation; however, we decided to not pursue these experiments because of the following reasons
>
> - For general-purpose generative models, it is quite hard to define a parametric form for the likelihood. In reality, for most data and modalities, assumptions of Gaussian or Mixture-of-Gaussian parameterization are quite limiting especially in high dimensions.
> - Explicit models offer a good alternative, but given their suboptimality in prediction problems we decided to explore generative modeling only for the framework that worked the best for us, which was implicit amortization.
>
> We agree that exploring various amortizations for generative modeling, different architecture choices, as well as model sizes can provide further insights. However, since we aim to unify such a wide range of approaches across a very diverse set of tasks, we had to limit our scope while working with the compute limitations that we had, and we believe that this should provide the first steps to the community to build from and improve various aspects of all the amortizations we considered.
>
> **Explanation to Fig 1**: We have instead added an explanation at the end of Section 2 and 4 to facilitate ease of understanding.
>
> **Likelihood**: We thank the reviewer for bringing this to our attention. We have added an explanation that provides a probabilistic interpretation of the framework, within which we then outline that the likelihood refers to the likelihood of the prediction conditioned on the observation. This naturally outlines that in (a) parametric approach, the likelihood is fixed with its parameters inferred using $g_\varphi$, (b) explicit approach, the likelihood family is fixed - indexed by $\gamma$ - with its partial parameters - aka latents - inferred using $g_\varphi$, while the (c) implicit approach does not offer a clean decomposition on the likelihood and instead directly models the posterior predictive.
>
> (1/2)

---

> ### Author Response · Authors · 2025-11-18
>
> **Causal Structure**: We use a standard causal transformer, which applies causal masking to the attention matrix. It is only for the implicit case where we need to augment this framework, since if we are learning to predict for $x_1, ... x_t$ conditioned on context $\mathcal{D}$, then using a causal transformer introduces dependencies between different test points which should not be there given the iid nature of the problem. Simply zeroing out those connections is also not enough since this would imply that we are only training for a single dataset size, or have to change the dataset size dynamically. Instead, our solution is more effective since it basically allows $x_i$ to look back at only a subset of the training dataset, allowing a parallelized version which effectively averages learning signal across different dataset sizes. This is illustrated in detail in Appendix E with a concise overview provided in Figure 6.
>
> **Comparison to Existing Methods**: We actually do compare to a number of existing methods. For parametric amortization the single step method with only data is equivalent to the known $z$ setup in [1]. We then use the same architecture to extend to optionally leverage gradient information (note that we keep the architecture as close as possible to enable comparisons without it acting as a confounder). For the explicit framework, leveraging only data with single step is equivalent to the explicit (bottleneck) model in [1,2] as well as conditional neural process [4] where DeepSets are replaced by transformers. Finally, the single step version for implicit amortization is equivalent to the implicit (non bottleneck) model in [1,2] as well as the architecture that is proposed in [5-8]. We have added Table 7 to convey related approaches and baselines more succinctly.
>
> **Errors in Table 2**: The parametric approach requires one to fully define a likelihood model, or fully define the form for prediction such that $g_\varphi$ is tasked with inferring all its parameters. This has two problems, (a) the form for the likelihood / predictor can be limiting even if $g_\varphi$ infers the optimal parameters for it, or (b) the form for the likelihood / predictor becomes quite complex such that inferring its parameters becomes increasingly difficult. In this work, for parametric approaches we limit our analysis to a linear predictor, which is suboptimal for complex tasks like ImageNet.
>
> We hope that our response addresses the reviewer's concerns and would be happy to answer any further questions.
>
> *References*
>
> [1] Mittal, Sarthak, et al. "Does learning the right latent variables necessarily improve in-context learning?." arXiv preprint arXiv:2405.19162 (2024).
>
> [2] Elmoznino, Eric, et al. "In-context learning and Occam's razor." arXiv preprint arXiv:2410.14086 (2024).
>
> [3] Mittal, Sarthak, et al. "Amortized in-context bayesian posterior estimation." arXiv preprint arXiv:2502.06601 (2025).
>
> [4] Garnelo, Marta, et al. "Conditional neural processes." International conference on machine learning. PMLR, 2018.
>
> [5] Garg, Shivam, et al. "What can transformers learn in-context? a case study of simple function classes." Advances in neural information processing systems 35 (2022): 30583-30598.
>
> [6] Von Oswald, Johannes, et al. "Transformers learn in-context by gradient descent." International Conference on Machine Learning. PMLR, 2023.
>
> [7] Müller, Samuel, et al. "Transformers can do bayesian inference." arXiv preprint arXiv:2112.10510 (2021).
>
> [8] Kirsch, Louis, et al. "General-purpose in-context learning by meta-learning transformers." arXiv preprint arXiv:2212.04458 (2022).
>
> (2/2)

---

> > ### Author Response · Authors · 2025-11-27
> > **Request for Discussion**
> >
> > Since we are only a week away from the end of the discussion phase, we would like to request the reviewer to kindly let us know if our response has adequately addressed their concerns. We would be happy to answer any further questions that the reviewer may have.

---

### Official Review · Reviewer_Z4Ku · 2025-10-21

**Soundness:** 3
**Presentation:** 4
**Contribution:** 2
**Rating:** 4
**Confidence:** 4

**Summary:**

This paper presents a unified framework for amortized learning, aiming to connect and clarify the relationship between methods like meta-learning, in-context learning (ICL), and learned optimizers. The authors propose that these diverse approaches can be understood through a common functional decomposition involving a task-adaptation function and a prediction function. Building on this framework, they introduce a taxonomy that categorizes models into three regimes: parametric, implicit, and explicit amortization, based on how they encode shared inductive biases and adapt to new tasks.
The paper also identifies a key weakness in existing methods: their limited ability to scale to large datasets during inference due to constraints like context length. To overcome this, the authors introduce Iterative Amortized Inference, a novel technique that refines a solution iteratively over mini-batches of task data.

**Strengths:**

1. The conceptual framework and taxonomy is nice and reasonable. The f(x, g(D_T)) formulation provides a clear and powerful lens for understanding the underlying mechanics of ICL, learned optimizers, and meta-learners, highlighting their shared principles.
The proposed taxonomy of parametric, implicit, and explicit amortization is intuitive and effectively organizes the space of amortized learning models.

2. Novel and well-motivated method. Iterative Amortized Inference is a novel solution to the critical problem of scaling in-context adaptation to large datasets. The idea of processing data in mini-batches at inference time is a natural and scalable extension of existing paradigms, merging the strengths of forward-pass and optimization-based approaches.

**Weaknesses:**

1. The unification of ICL and learned optimizers, and the taxonomy, are not novel. ICL has been viewed as a form of meta-learners in literature [1,2,4].
And the f(x, g(D_T)) framework also has exists in literature for a not short time [1,3].
Especially, [1] has unified ICL and meta-learners explicitly and theoretically, under the same f(x, g(D_T)) framework. As learned optimizer is a subset of meta-learners, this paper exactly falls into the existing paradigm. However,  [1] has not even been referred, as a very closely related work.

2. Limitation of proposed method:
the baseline comparison and application scenario discussion are insufficient. Note that most experiments follows meta-learning setting, where numerous meta-learning methods are feasible, but this paper has only compared with 1-step amortization with transformer (e.g., standard ICL). Even restricting in this scope, it seems that training set size per task would play a critical role in the effect of iterating over one-step, which should be ablation studied and discussed. Existing results are far from to be able to see when and what advantage IAI would bring comparing with feasible baselines in practice.

3. Some technical design of IAI is questioned. Please refer to Questions 1.

4. The final implementation of IAI has hardly distinct with existing auto-regressive training, which is a common practice in ICL with transformer. Please refer to Question 2.

[1] Why In-Context Learning Models are Good Few-Shot Learners?. ICLR2025

[2] General-purpose in-context learning by meta-learning transformers. arXiv:2212.04458

[3] Conditional neural processes. ICML 2018. and Neural Processes. ICML 2018 workshop

[4] Learning to Learn with Contrastive Meta-Objective. arXiv:2410.05975

**Questions:**

1. For parametric and explicit setup, the authors treat $\theta$ as a token to be input to transformer, along with other tokens which represent context. Considering the possible very large discrepancy in dimensions (e.g.,$ W_T \in R^{784\*100} $) while context units is about 8*8 in MNIST), how to feed $\theta$ as an input token to transformer? And how to unify different modalities of input tokens (parameter, data, gradient)?
The reviewer think the traditional way to modulate $\theta$ by some functions using the output of $h_\psi$ [5], rather than feeding $\theta$ into transformer, might be more reasonable and practical.

2. The principle idea of introducing iteration into inference, and the implementation, seems to be similar to auto-regressive training that has been commonly applied in ICL with transformer. What is the difference?

3. The usage and meaning of  "amortization" seems different from literature [6]. I am curious where this term originates from, and why this paper use it in this way.

[5] Fast and flexible multi-task classification using conditional neural adaptive processes. NIPS2019

[6] Memory efficient meta-learning with large images. NIPS2021

---

> ### Author Response · Authors · 2025-11-18
>
> We thank the reviewer for their useful feedback and have updated the manuscript with their suggestions. We hope that our response addresses the concerns raised by the reviewer.
>
> **Novelty and connection to [1]**: Our contribution is not to claim that ICL is a form of meta-learning as this has indeed been established in a number of prior works [1,3,7–12]. Instead, our contribution is to show that a broad class of meta-learners (not just ICL, but also neural processes, hypernetworks, and others) can be mathematically unified through Equation (5), which provides an algorithmic decomposition that makes the information flow explicit. This perspective highlights how the conditioning dataset influences the downstream prediction through the interaction of $f_\gamma$ and $g_\varphi$, enabling us to introduce a clear and interpretable dichotomy between parametric, explicit, and implicit meta-learners. To our knowledge, this dichotomy has never been outlined in the literature. Further, we present a novel use of this dichotomy's components in the form of IAI.
>
> We thank the reviewer for pointing us to [1]. While related, our contribution differs substantially from theirs, and we have now added a discussion of this work in the revised draft in the appendix. [1] provides a theoretical expressivity analysis showing that ICL can model several forms of meta-learning solutions, but it does so largely through universal approximation arguments. Their analysis does not account for information flow, conditional independence structure, or the properties of the conditioning dataset. However, properties of the dataset and parameterizations of $f_\gamma$ and $g_\varphi$ play an important role in the optimization dynamics and modeling capabilities of different meta-learners, especially in finite-capacity regimes where universal approximation does not apply.
>
> Finally, neither [1] nor [3] present a general formulation of meta-learning in terms of $f_\gamma(x, g_\varphi(\mathcal{D}))$. Instead, they analyze specific instantiations of these functions, whereas our work develops a unified framework that applies across a much broader range of meta-learning methods.
>
> **Comparison to Baselines**: We actually do compare to a number of baselines [2,3,7-15], though sometimes we replace less performant backbone architectures like RNNs or DeepSets (eg. in [2]) with Transformers to allow for a fairer comparison. We present the baselines used on page 8, as well as include these details in the new Table 7. Following are the baselines we consider
>
> - Table 2, Grad row (learned optimizers)
> - Table 2, Data row with 1-step (hypernetworks; [13] with RNN replaced with Transformers, known predictor model in [7], and point estimator in [14])
> - Table 3, Data row with 1-step (conditional neural process; [2], explicit model in [7-8])
> - Table 4-5, row with 1-step (ICL; [3,10-12] and the implicit model in [7-8])
>
> We have included both a concise summary of the different amortizations as well as the baselines considered in Table 7.
>
> **Training set size and advantage of IAI**: We train all the meta-learners on the same set of training tasks that are continually randomly sampled, as outlined in Appendix C. The primary benefit of IAI is that it disentangles the dataset size from context length, allowing the meta-learner to take into account more observations than what is constrained by its context length. This is both discussed and assessed with the baselines outlined above, with details about the meta-training distribution in Appendix C. For each individual task, we consider a dataset size of $1000$ from which mini-batches are randomly sampled with $100$ observations.
>
> **Discrepancy in dimensions**: We currently use separate linear layers for the different modalities (inputs, states, gradients), as outlined in Appendix E.1, which is standard practice when using inputs from multiple modalities in a shared architecture. The reviewer has a good point regarding [5]. We tried to keep the comparisons fair by fixing an architecture without additional independence assumptions (eg. [5] models per-class weights as independent of other classes), but we do believe that specific extensions into various sub-categories that we provided is very relevant future direction (what the reviewer is saying falls into a specific parameterization of the explicit model).
>
> **Comparison to auto-regressive training**: Our framework is fundamentally different from auto-regressive training since at each step, we predict an updated state based on the mini-batch which is then fed back into the model, as opposed to auto-regressive training which feeds the entire sequence at once.
>
> (1/2)

---

> > ### Author Response · Authors · 2025-11-18
> >
> > Autoregressive tranformer models (eg. LLMs) can be trained through direct supervision since they model observed data and can thus enjoy teacher forcing during training. In our case, there are no ground-truth states to derive direct supervision from, which makes it a fundamentally different problem. The following examples should clarify the differences:
> >
> > - Learned optimizers: These frameworks iteratively update parameters of the model. However, they cannot be trained like LLMs because we don't have access to a "true" trajectory of parameters. We can obtain a trajectory by using our favourite optimizer but then we would just learn to mimic that optimizer, but the goal of learned optimizers is to outperform them. There is, however, indirect supervision; given a parameter we can evaluate how good it is through downstream prediction on the data that we have.
> > - Hierarchical variational inference: Denotes a similar problem, where we don't have ground-truth data about the latent variables. Instead, we have indirect supervision about how good these latent variables are by seeing how well they explain the data.
> >
> > **Term amortization**: It is used in the same sense in both our work and [6], though [6] uses it in a much narrower sense than us. Amortization refers to the ability to reduce future costs through efficient conditional computations. For example, in VAEs instead of learning the posterior over latents from scratch for each observation, a shared conditional encoder $q(z|x)$ is trained. Similarly, instead of training a new model from scratch for new data, hypernetworks / ICL / MAML / etc. are used since they amortize the cost of all future new task trainings into a more efficient and cheaper model that can solve the inference problem for new datasets more cheaply over time (through conditioning on the new dataset).
> >
> > We thank the reviewer for pointing us to some of the recent literature and have updated our draft to include them. We hope that our response addresses the reviewer's concerns and would be happy to answer any further questions.
> >
> > *References*
> >
> > [1] Why In-Context Learning Models are Good Few-Shot Learners?. ICLR2025
> >
> > [2] Garnelo, Marta, et al. "Conditional neural processes." International conference on machine learning. PMLR, 2018.
> >
> > [3] Kirsch, Louis, et al. "General-purpose in-context learning by meta-learning transformers." arXiv preprint arXiv:2212.04458 (2022).
> >
> > [4] Learning to Learn with Contrastive Meta-Objective. arXiv:2410.05975
> >
> > [5] Fast and flexible multi-task classification using conditional neural adaptive processes. NIPS2019
> >
> > [6] Memory efficient meta-learning with large images. NIPS2021
> >
> > [7] Mittal, Sarthak, et al. "Does learning the right latent variables necessarily improve in-context learning?." arXiv preprint arXiv:2405.19162 (2024).
> >
> > [8] Elmoznino, Eric, et al. "In-context learning and Occam's razor." arXiv preprint arXiv:2410.14086 (2024).
> >
> > [9] Mittal, Sarthak, et al. "Amortized in-context bayesian posterior estimation." arXiv preprint arXiv:2502.06601 (2025).
> >
> > [10] Garg, Shivam, et al. "What can transformers learn in-context? a case study of simple function classes." Advances in neural information processing systems 35 (2022): 30583-30598.
> >
> > [11] Von Oswald, Johannes, et al. "Transformers learn in-context by gradient descent." International Conference on Machine Learning. PMLR, 2023.
> >
> > [12] Müller, Samuel, et al. "Transformers can do bayesian inference." arXiv preprint arXiv:2112.10510 (2021).
> >
> > [13] Ha, David, Andrew Dai, and Quoc V. Le. "Hypernetworks." arXiv preprint arXiv:1609.09106 (2016).
> >
> > [14] Mittal, Sarthak, et al. "In-Context Parametric Inference: Point or Distribution Estimators?." arXiv preprint arXiv:2502.11617 (2025).
> >
> > [15] Reuter, Arik, et al. "Can Transformers Learn Full Bayesian Inference in Context?." arXiv preprint arXiv:2501.16825 (2025).
> >
> > (2/2)

---

> > > ### Author Response · Authors · 2025-11-27
> > > **Request for Discussion**
> > >
> > > Since we are only a week away from the end of the discussion phase, we would like to request the reviewer to kindly let us know if our response has adequately addressed their concerns. We would be happy to answer any further questions that the reviewer may have.

---

> > > > ### Comment · Reviewer_Z4Ku · 2025-11-28
> > > >
> > > > Thanks for the authors' feedback. Some of my concerns has been addressed. I have raised my rating. I think most part of this paper is well-motivated and organized. However,  from my perspective, still, the proposed IAI is inherently the same with, or at least, both the training and inference processes can be expressed by auto-regressively trained ICL transformer, which is existing.

---

### Official Review · Reviewer_8sW6 · 2025-10-21

**Soundness:** 4
**Presentation:** 4
**Contribution:** 3
**Rating:** 8
**Confidence:** 4

**Summary:**

This paper proposes Iterative Amortized Inference (IAI), a unified framework that generalizes amortized learning approaches (meta-learning, ICL, prompt tuning and learned optimizers) under a common mathematical framework. Furthermore, the authors provide a categorization of amortized models into parametric, implicit, and explicit regimes. Finally, they also propose a mechanism inspired by stochastic optimization, which processes mini-batches of task data incrementally.

**Strengths:**

The unified perspective proposed in the paper is valuable, as it clearly articulates how different amortized learning methods relate to one another within a common framework. The taxonomy provides a useful conceptual structure that can guide future research in this area. Moreover, the proposed IAI approach effectively addresses the scalability limitations of previous methods. The empirical results are convincing, demonstrating consistent improvements over existing baselines in both in-distribution and out-of-distribution settings.

**Weaknesses:**

- The paper could benefit from more general reference to the meta-learning literature. Including references to survey papers in the main text could help the reader to better understanding the meta-learning problem, which is only detailed in the appendix.
- While Section 5 discusses memory usage and computational cost, the analysis lacks details on the runtime required to train a k-step iterative model.
- It would be valuable to discuss theoretical guarantees (e.g., convergence bounds) for IAI.
- Minor issue: it would be helpful to illustrate $g_\psi$ in Figure 1.

**Questions:**

- The paper mentions improved scalability through iterative amortization. Could the authors clarify whether this scalability refers primarily to model size, dataset size, task complexity, or another dimension? A more detailed evaluation along the chosen dimension would strengthen the claim.
- How would recent works on meta-learning with in-context learning (e.g., [1,2]) fit into the proposed taxonomy?
- The experimental tables report error rates rather than average accuracy. Is there a specific reason for this choice? Reporting accuracy would make it easier to compare with previous studies.
- How is the number of steps for IAI selected? Do additional steps continue to yield improvements, or is there a point of diminishing returns?
- Given the iterative refinement process, IAI appears well suited for continual or lifelong learning setups. Have the authors considered this application, or do they foresee challenges in extending IAI to such scenarios?

---

> ### Author Response · Authors · 2025-11-18
>
> We thank the reviewer for their useful feedback and have updated the manuscript with their suggestions. We hope that our response addresses the concerns raised by the reviewer.
>
> **General references to meta-learning literature**: We have added more references in the introduction, and would be happy to include additional references to provide more context about the framework. Are there specific surveys that the reviewer is referring to?
>
> **Training runtime**: We thank the reviewer for the thoughtful comment. Under the greedy learning framework described in Section 4, training cost scales linearly with the number of steps. In our experiments, we train all models to convergence. However, in practice the effective cost is often sublinear: parameter updates that improve performance at step $i$ can also improve performance at step $j$. This interaction makes it difficult to precisely characterize the required training runtime, since the training signals across different steps are not independent and can generalize to one another.
>
> **Theoretical guarantees**: We fully agree with the reviewer. While our work provides an initial step toward unifying various meta-learners and extending them to an iterative framework, a deeper analysis of their convergence properties would indeed be important and valuable. Unfortunately, developing such results is challenging because a general theory of meta-learning and its convergence behavior is still largely missing. Nonetheless, we view this as a highly relevant and promising direction for future work.
>
> > Minor issue: it would be helpful to illustrate in Figure 1.
>
> Could the reviewer clarify what they mean?
>
> **Axis for scalability**: We refer to the dataset size when we talk about scalability; highlighting that in-context learners are limited by their context length which is exactly the limitation that we aim to tackle.
>
> **Recent works on meta-learning with in-context learning**: Could the reviewer tell us which papers they are referring to?
>
> **Error rates**: We chose to report error rates to maintain consistency with our regression and generative metrics, where lower values indicate better performance. We felt that this made the comparisons cleaner and more uniform.
>
> **Number of steps**: For now, we just consider the number of steps as a hyperparameter. Our experiments consistently highlight that there are diminishing returns with increasing steps, as can be seen from the gap in performance between $1\to 5$ steps and $5\to 10$ steps.
>
> **Continual / Lifelong learning**: This is an excellent point and we believe a very relevant future direction. We do foresee a challenge that such an amortized learner would need to be meta-trained on somewhat similar flavours of continually updating task structures, otherwise the meta-learner may not generalize at inference. It would be great to have meta-datasets with such temporal patterns, and then extending IAI to them.
>
> We hope that our response addresses the reviewer's concerns and would be happy to answer any further questions.

---

> ### Comment · Reviewer_8sW6 · 2025-11-19
>
> Thank you for the detailed responses to my comments. I have a few follow-ups clarifications:
>
> **Minor issue: representation in Figure 1.** Since Figure 1 shows the difference between explicit and implicit amortized models, I was wondering whether it would be possible to represents $g_\psi$ in the figure. This would make the difference more explicit as $g_\psi$ is learnable in the explicit models, and fixed in the implicit models.
>
> **Recent works on meta-learning with in-context learning**. My apologies for not providing the references earlier. The works I was referring to are: *[1] Fifty, Christopher, et al. "Context-aware meta-learning." arXiv preprint arXiv:2310.10971 (2023)* and *[2] Vettoruzzo, Anna, et al. "Unsupervised meta-learning via in-context learning." arXiv preprint arXiv:2405.16124 (2024).* and *[3] General-Purpose In-Context Learning by Meta-Learning Transformers https://arxiv.org/abs/2212.04458* . It would be interesting to see how these approaches fit into your proposed taxonomy.
>
> **Error rates**. Thank you for the clarification. To avoid possible ambiguity for readers, I would recommend adding a short note in the table captions clarifying that lower values are better, since the error rate is not the standard metric used for classification tasks.
>
> **Number of steps**. From tables 4,5,6 it seems that 10 steps gives the best performance. It would be useful to know whether increasing the number of steps beyond 10 yields continued improvements or whether performance plateaus. Even a brief discussion or an additional small-scale experiment could clarify this behavior.
>
> **Continual / Lifelong learning** I think the iterative design of your method naturally suggests potential applications to continual or lifelong learning. Including this in the future work could highlight an interesting direction for follow-up research.

---

> > ### Author Response · Authors · 2025-11-25
> >
> > We thank the reviewer for their response.
> >
> > **Figure 1**: Thanks for clarifying this! We have updated the draft to highlight what parts of the system are impacted with $g_\varphi$.
> >
> > **References**: Thanks for bringing these works to our attention. They fall within the domain of the implicit model, in particular [3] is exactly the 1-step process that we use in Tables 4-5. We have updated the draft to cite [1-2]. Notably, none of [1-3] reduce the dataset into a compressed representation -- i.e. no consolidation of the entire dataset into low-dimensional latents -- and instead they let the query attend to each observation through a learned system.
> >
> > **Error Rates**: We have updated the draft to explicitly point out that lower is better across all metrics.
> >
> > **Number of Steps**: Thanks to the reviewer, we have run an additional experiment on MNIST classification setting. In particular, we train a single model to perform classification for up to 15 steps and showcase its performance below, again with lower as better (error metric).
> >
> > | Dataset | Step 1 | Step 5 | Step 10 |  Step 15 |  Step 20 |
> > |-----------|----------:|----------:|----------:| ----------:|----------:|
> > | MNIST |    23.3%       | 11.8% |     9.5%      |     8.8%      |   **8.5%** |
> > | FMNIST    |    31.8%       |     25.3%     |  24.0  |  **23.8%**      | 23.9% |
> >
> > We first see that there is a slight improvement in performance when going from 10 to 15 steps both ID and OoD. However, going from 15 to 20 steps, which is OoD in number of steps, there is slight improvement on the ID dataset and slight degradation in the OoD dataset.
> >
> > **Continual / Lifelong Learning**: We have updated the draft and added a discussion about how continual / lifelong learning are important future directions.
> >
> > We hope that the additional experiment and updates to the draft have resolved the reviewer's concerns.
> >
> > [1] Fifty, Christopher, et al. "Context-aware meta-learning." arXiv preprint arXiv:2310.10971 (2023)
> >
> > [2] Vettoruzzo, Anna, et al. "Unsupervised meta-learning via in-context learning." arXiv preprint arXiv:2405.16124 (2024).
> >
> > [3] General-Purpose In-Context Learning by Meta-Learning Transformers https://arxiv.org/abs/2212.04458

---

> > > ### Comment · Reviewer_8sW6 · 2025-11-26
> > >
> > > I appreciate the authors' clarifications and I confirm my score, as the paper presents a solid and valuable contribution.

---

> > > > ### Author Response · Authors · 2025-11-26
> > > >
> > > > We thank the reviewer for engaging in discussion and are happy that they find value in the work!

---

### Official Review · Reviewer_EnCU · 2025-10-28

**Soundness:** 3
**Presentation:** 2
**Contribution:** 2
**Rating:** 4
**Confidence:** 3

**Summary:**

In this paper, the authors present a unified framework that encompasses various amortized learning paradigms, including meta-learning, context learning and learned optimization. Within this framework, a learned function $ f_{\gamma}$ operates in conjunction with another function $ g_{\phi}$. Specifically, $f_{\gamma}$ performs inference based on the knowledge that $g_{\phi}$ has acquired from a dataset $ D_{t}$. This formulation integrates the key characteristics of different amortized methods, allowing each individual approach to be viewed as a special case}within the broader framework. Furthermore, the authors categorize tasks into three classes, parametric, implicit, and explicit, which help distinguish among different amortized models according to whether $f_{\gamma}$ and $g_{\phi}$  are trainable or fixed. In addition, they propose a scalable sequential inference mechanism, where the output from each step serves as the input to the next, effectively modeling the process as a sequence-based system.

**Strengths:**

1: This paper investigates the similarities among different amortized methods, which are typically treated as distinct research areas. The conclusions drawn from this unified analysis are meaningful, as they help reveal the common underlying principles shared across these approaches.

2: The authors categorize various amortized methods based on whether the functions f and g are trainable or fixed. This categorization offers an interesting and novel perspective for understanding the relationships between different amortized learning paradigms.

3: The authors provide sufficient theoretical context and analysis to support the effectiveness of the proposed framework, thereby strengthening its conceptual soundness and empirical credibility.

**Weaknesses:**

1: The presentation of the paper is somewhat confusing. It lacks a clear clarification of the unified framework or an algorithmic summary. The purely descriptive explanations make it difficult to understand how the proposed unified system actually works in practice.

2: The explanation of the iterative amortized inference process is unclear. The distinction between the proposed method and existing approaches such as meta-learning is vague, which makes it challenging to identify the novel contributions of this work.

3: Since the framework aims to unify different amortized methods, the authors should provide a more thorough comparison with existing approaches across multiple baselines. This would help ensure that the unification does not lead to a significant performance drop relative to specialized methods.

**Questions:**

1: It would be helpful to include a concise summary of the methods and an algorithmic description, especially for the iterative amortized inference component. Because the framework spans multiple cases, a step-by-step outline (inputs, updates, outputs) would make the overall system easier to follow.

2: A more thorough comparison against baselines from multiple categories (e.g., meta-learning, context learning, learned optimizers) would better demonstrate the framework’s effectiveness. Reporting results across diverse, representative baselines would clarify where the unified approach helps and where specialized methods still have an edge.

---

> ### Author Response · Authors · 2025-11-18
>
> We thank the reviewer for their useful feedback and have updated the manuscript with their suggestions. We hope that our response addresses the concerns raised by the reviewer.
>
> **Clarification of the unified framework**: We thank the reviewer for pushing us to improve clarity as a central goal of this work is to clearly draw parallels between related methods. Section 2 is dedicated to algorithmic description of the different components ultimately captured by a central expression found in Equation (5). While this section along with the unified Equation (5) were present in the original draft, we have improved their presentation. Notably, we now explicitly point to them following the descriptive explanations in the introduction. Furthermore, we have added a paragraph further unpacking the unified framework presented in Equation (5) immediately following its presentation (explanation found on page 4).
>
> For clarity and to summarize, the unified framework neatly describes various amortized inference procedures into (a) an optional bottleneck $g_\varphi$, and (b) a downstream predictor $f_\gamma$, such that different choices encompass different meta learners. Specific examples of different choices are mathematically outlined in Table 1 and Section 2.1. We hope that our response and the added explanation in the manuscript clarifies how the unified framework works.
>
> **Explanation of Iterative Amortized Inference**: Following up on the above explanation, iterative amortized inference *extends* existing meta-learning frameworks to a recurrent $g_\varphi$ and/or $f_\gamma$. The presentation of these components is now improved.
>
> About novelty, to the best of our knowledge, meta learners only leverage an iterative framework with learned optimizers which operate solely based on gradients. IAI's novelty lies in extending various other meta learners -
>
> - Parametric amortization extends the known predictor framework of [1] as well as the hypernetworks framework [3,9-11].
> - Within explicit amortization, we extend the explicit (bottleneck) model in [1,2] and the conditional neural process [4], repurposed with a transformer backbone.
> - In implicit amortization, we extend the implicit model of [1,2] as well as [5-8].
> - Finally, we additionally extend the ICL framework to generative modeling.
>
> Appendix B additionally provides a detailed walk-through of the overall pipeline and we also provide an intuitive explanation of IAI on page 7. Since our framework subsumes a wide variety of meta-learning methods from MAML to ICL, we had to maintain a level of abstractness to be as general as possible but we hope the specifics provided in our response clarifies the reviewer's concerns. In summary, we argue that our manuscript introduces (1) a novel unified comparison of several methods that share algorithmic components and (2) the presentation of a novel method that exploits new combinations of these components.
>
> **Comparison to Baselines**: We agree with the reviewer that baseline comparison is key in this type of study. We note that we made a concerted effort to provide such comparisons across a number of widely used baselines [1-11]. For fair comparison, we note that we sometimes replace less performant backbone architectures like RNNs or DeepSets (eg. in [4]) with Transformers to allow for a fairer comparison. We present the baselines methods on page 8, as well as include these details in the new Table 7. Following are the baselines we consider
>
> - Table 2, Grad row (learned optimizers)
> - Table 2, Data row with 1-step (hypernetworks; [9] with RNN replaced with Transformers, known predictor model in [1], and point estimator in [10])
> - Table 3, Data row with 1-step (conditional neural process; [4], explicit model in [1-2])
> - Table 4-5, row with 1-step (ICL; [5-8] and the implicit model in [1-2])
>
> We have included both a concise summary of the different amortizations as well as the baselines considered in Table 7. We hope that our response addresses the reviewer's concerns and would be happy to answer any further questions.
>
> (1/2)

---

> > ### Author Response · Authors · 2025-11-18
> >
> > *References*
> >
> > [1] Mittal, Sarthak, et al. "Does learning the right latent variables necessarily improve in-context learning?." arXiv preprint arXiv:2405.19162 (2024).
> >
> > [2] Elmoznino, Eric, et al. "In-context learning and Occam's razor." arXiv preprint arXiv:2410.14086 (2024).
> >
> > [3] Mittal, Sarthak, et al. "Amortized in-context bayesian posterior estimation." arXiv preprint arXiv:2502.06601 (2025).
> >
> > [4] Garnelo, Marta, et al. "Conditional neural processes." International conference on machine learning. PMLR, 2018.
> >
> > [5] Garg, Shivam, et al. "What can transformers learn in-context? a case study of simple function classes." Advances in neural information processing systems 35 (2022): 30583-30598.
> >
> > [6] Von Oswald, Johannes, et al. "Transformers learn in-context by gradient descent." International Conference on Machine Learning. PMLR, 2023.
> >
> > [7] Müller, Samuel, et al. "Transformers can do bayesian inference." arXiv preprint arXiv:2112.10510 (2021).
> >
> > [8] Kirsch, Louis, et al. "General-purpose in-context learning by meta-learning transformers." arXiv preprint arXiv:2212.04458 (2022).
> >
> > [9] Ha, David, Andrew Dai, and Quoc V. Le. "Hypernetworks." arXiv preprint arXiv:1609.09106 (2016).
> >
> > [10] Mittal, Sarthak, et al. "In-Context Parametric Inference: Point or Distribution Estimators?." arXiv preprint arXiv:2502.11617 (2025).
> >
> > [11] Reuter, Arik, et al. "Can Transformers Learn Full Bayesian Inference in Context?." arXiv preprint arXiv:2501.16825 (2025).
> >
> > (2/2)

---

> ### Author Response · Authors · 2025-11-27
> **Request for Discussion**
>
> Since we are only a week away from the end of the discussion phase, we would like to request the reviewer to kindly let us know if our response has adequately addressed their concerns. We would be happy to answer any further questions that the reviewer may have.

---

### Meta-Review · Area_Chair_EPaE · 2026-01-07

**Summary:**

- All reviewers think the experimental results are insufficient, lacking comparison with existing baselines, only considering limited scenarios (EnCU and Z4Ku), and only one network architecture is considered (Reviewer LiiH), and no runtime is provided (8sW6). Even though the rebuttal provided additional runtime table, they didn't provide experiments for additional tasks or baselines, or other architectures.

- The majority of the reviewers finds the paper's presentation confusing, there are insufficient details about the framework, implementation, experiment settings, and numerical results, making the paper unclear.

- Several reviewers point out the insufficient literature review and comparison with existing frameworks.

- After discussion, Reviewer Z4Ku still thinks the novelty in the proposed method is rather limited due to the similarity between the proposed method and the existing method.

- After reading the rebuttal, I think the paper could benefit from a major revision by adding more experimental results and more explanation, and improving the presentation significantly.

**Reviewer Concerns:**

**Addressed Concerns**

- Reviewer EnCu's concern on insufficient literature review is addressed successfully in the rebuttal.

- Reviewer 8sW6's concern on runtime is addressed by additional experiment result in the rebuttal.

- Reviewer 8sW6 asks several clarification questions, e.g. definitions of scalability in this paper, literature comparison, numerical results, and implementation details. The rebuttal clarified them successfully.

- Reviewer LiiH also asked several clarification questions, which were successfully addressed in the rebuttal.

**Remaining Concerns**
- The concerns on insufficient experiment results, especially baselines and tasks, are not addressed completely. The rebuttal only re-emphasizes the existing results in the paper.

- The presentation of the paper needs a more thorough revision, as pointed by multiple reviewers.

- The major concern of Reviewer Z4Ku is still unaddressed after discussion, as pointed out by the reviewer themselves. This is about the novelty of the proposed method due to the similarity between the proposed IAI and the existing ICL method with transformer.

**Reviewer Scores:**

Reviewer EnCu (score 4) is not likely to change the score.

Reviewer 8sW6 (score 8) and Reviewer (LiiH) may decrease their score.

Reviewer Z4Ku said they will increase their score from score 4.

---

### Decision · Program_Chairs · 2026-01-26

Reject